# Actomyosin drives cancer cell nuclear dysmorphia and threatens genome stability

Tohru Takaki[1,2], Marco Montagner[3,*], Murielle P. Serres[1,4,*], Maël Le Berre[5], Matt Russell[6], Lucy Collinson[6], Karoly Szuhai[7], Michael Howell[8], Simon J. Boulton[2], Erik Sahai[3] & Mark Petronczki[1,9]

Altered nuclear shape is a defining feature of cancer cells. The mechanisms underlying nuclear dysmorphia in cancer remain poorly understood. Here we identify PPP1R12A and PPP1CB, two subunits of the myosin phosphatase complex that antagonizes actomyosin contractility, as proteins safeguarding nuclear integrity. Loss of PPP1R12A or PPP1CB causes nuclear fragmentation, nuclear envelope rupture, nuclear compartment breakdown and genome instability. Pharmacological or genetic inhibition of actomyosin contractility restores nuclear architecture and genome integrity in cells lacking PPP1R12A or PPP1CB. We detect actin filaments at nuclear envelope rupture sites and define the Rho-ROCK pathway as the driver of nuclear damage. Lamin A protects nuclei from the impact of actomyosin activity. Blocking contractility increases nuclear circularity in cultured cancer cells and suppresses deformations of xenograft nuclei *in vivo*. We conclude that actomyosin contractility is a major determinant of nuclear shape and that unrestrained contractility causes nuclear dysmorphia, nuclear envelope rupture and genome instability.

[1] Cell Division and Aneuploidy Laboratory, Cancer Research UK London Research Institute, Clare Hall Laboratories, South Mimms, Hertfordshire EN6 3LD, UK. [2] DSB Repair Metabolism Laboratory, The Francis Crick Institute, 1 Midland Road, London NW1 1AT, UK. [3] Tumour Cell Biology Laboratory, The Francis Crick Institute, 1 Midland Road, London NW1 1AT, UK. [4] MRC Laboratory for Molecular Cell Biology, UCL, Gower Street, London WC1E 6BT, UK. [5] Institut Curie, PSL Research University, CNRS, UMR 144, F-75005 Paris, France. [6] Electron Microscopy Group, The Francis Crick Institute, 1 Midland Road, London NW1 1AT, UK. [7] Department of Molecular Cell Biology, LUMC, Einthovenweg 20, 2333 ZC Leiden, The Netherlands. [8] High Throughput Screening Laboratory, The Francis Crick Institute, 1 Midland Road, London NW1 1AT, UK. [9] Boehringer Ingelheim RCV GmbH & Co KG, Dr Boehringer Gasse 5-11, A-1121 Vienna, Austria. * These authors contributed equally to this work. Correspondence and requests for materials should be addressed to E.S. (email: erik.sahai@crick.ac.uk) or to M.P. (email: mark_paul.petronczki@boehringer-ingelheim.com).

Alterations in nuclear shape and size as well as ongoing genomic changes are defining features of cancer cells and are used to clinically grade malignancies. The physical properties of the nucleus are largely determined by the chromatin contained within the nucleus and the nuclear envelope. The nuclear envelope is composed of two lipid bilayer membranes, nuclear pores spanning the two membranes and a network of nuclear lamina proteins, including LMNA, LMNB1 and LAP2β, associated with the inner membrane[1,2]. Disruption of LMNA and LMNB1 function leads to altered nuclear morphology and gene expression[3]. The cytoplasmic face of the nuclear envelope can be coupled to the cytoskeleton and this linkage may influence nuclear architecture through mechanical forces[4]. However, the interplay between cytoskeletal forces and nuclear shape is not well understood. Although nuclear dysmorphia has been used for tumour grading in the clinic for several decades[5,6], the mechanisms underlying aberrant nuclear morphology in cancer and their impact on genomic integrity remain largely unaddressed[6,7].

In non-muscle cells, contractile force is predominantly generated by the action of myosin II motors on actin filaments[8]. Phosphorylation of myosin regulatory light chain (MYL9) on threonine 18 and serine 19 promotes productive actomyosin interactions and force generation[9,10]. The small G-proteins RhoA and RhoC are able to promote myosin regulatory light chain (also known as MRLC) phosphorylation by activating the kinases ROCK1 and ROCK2, and gain of Rho/ROCK function is linked to aggressive tumour phenotypes[11–13]. De-phosphorylation of myosin light chain is mediated by a protein phosphatase 1 (PP1) complex containing the regulatory subunit, PPP1R12A (also known as MYPT1)[14]. Thus, PPP1R12A normally functions to restrain actomyosin contractility.

We find that actomyosin contractility is major determinant of nuclear shape and integrity both *in vitro* and *in vivo*. We identify myosin phosphatase as a safeguard for nuclear integrity. Unrestrained contractility caused by loss of PPP1R12A elicits nuclear envelope rupture and drives genome instability. Our data suggest that nuclear dysmorphia of cancer cells and nuclear shape in general is controlled by the opposing activities of actomyosin contractility promoting factors and myosin phosphatase.

## Results

**Myosin phosphatase safeguards nuclear integrity**. In this study, we set out to identify regulators of nuclear morphology in human cells using functional genomics. We systematically perturbed protein phosphorylation by using RNA interference (RNAi) to deplete 264 phosphatase factors in HeLa cells (Supplementary Data 1). Following siRNA transfection, the nuclear morphology defects in cells were evaluated by microscopy (Supplementary Fig. 1a). The top hit emerging from this analysis was PPP1R12A (also called MYPT1, myosin phosphatase target subunit), which, when depleted, led to highly irregular and fragmented nuclei that diverged greatly from the simple circular form of control nuclei (Fig. 1a,b; Supplementary Fig. 1a,b and Supplementary Movies 1 and 2). PPP1R12A is a regulator of PPP1CB (also called PP1β), one of three enzymatic protein phosphatase 1 (PP1) genes encoded in the human genome. The regulatory subunit PPP1R12A directly binds to the catalytic subunit PPP1CB and is thought to guide substrate selectivity of the latter[15]. The complex of PPP1R12A and PPP1CB is also referred to as myosin phosphatase and has been reported to dephosphorylate multiple substrates including myosin light chain and the mitotic kinase PLK1 (ref. 16). Consistent with complex formation between the two PP1 subunits, PPP1CB depletion also caused a dramatic

change in nuclear morphology as well as nuclear fragmentation in our RNAi screen, similar to depletion of PPP1R12A (Fig. 1a,b; Supplementary Fig. 1a,b). Previous screens have also identified PPP1R12A and PPP1CB as factors controlling nuclear morphology[17,18]. However, the nature of the nuclear defect observed in depleted cells and the function of the two proteins in maintaining nuclear shape has not been addressed. Multiple independent siRNA duplexes targeting PPP1R12A or PPP1CB caused strong nuclear morphology defects (Supplementary Fig. 1c). Expression of siRNA-resistant transgenes encoding the phosphatase subunits PPP1R12A and PPP1CB restored normal nuclear shape upon depletion of the endogenous counterparts (Supplementary Fig. 2). These results demonstrate that the phosphatase enzyme PPP1CB and its regulator PPP1R12A are required to maintain normal nuclear morphology in cultured human cells. Co-depletion of both phosphatase subunits did neither suppress nor further enhance the nuclear defects observed by individual depletion of each subunit suggesting that the observed phenotype reflects a loss-of-function situation of the phosphatase complex (Supplementary Fig. 1d).

**Loss of PPP1R12A–PPP1CB causes nuclear lamina rupture**. To address the nature of the striking nuclear deformation caused by depletion of PPP1R12A and PPP1CB, we first focused on the nuclear envelope. Co-staining for nuclear envelope markers and DNA revealed that PPP1R12A and PPP1CB depletion led to discontinuities of the nuclear envelope and the emergence of DNA segments uncovered by the nuclear lamina (Fig. 1a; Supplementary Fig. 3a and Supplementary Movies 3,4,5). Remarkably, these segments of DNA in PPP1R12A or PPP1CB-depleted cells were not covered by any marker of the nuclear envelope tested, including lamin A and B1, the lamina-associated protein LAP2β, nuclear pore complex proteins and the inner nuclear membrane protein SUN1. Time-lapse imaging revealed that the nuclear envelope was highly dynamic and rapidly re-formed over areas of uncovered chromatin (Supplementary Movies 1,2). These results indicate a rupture of the nuclear envelope. The expression of the chromatin and nuclear envelope marker proteins necessary for time-lapse imaging (H2B-mCherry and AcGFP-LAP2β, respectively) did not alter nuclear shape in untreated cells or the nuclear morphology phenotype caused by depletion of PPP1R12A (Supplementary Fig. 4a,b). The majority of chromatin segments that were not covered by nuclear envelope markers in myosin phosphatase-depleted cells remained connected to the main nuclear mass through thin chromatin bridges indicating that these segments were derived from a single nucleus (Supplementary Fig. 3b). A minority of the extruded chromatin segments was no longer connected to main nuclear body. The nuclear integrity defects observed in PPP1R12A or PPP1CB-depleted cells differ from previously described nuclear envelope aberrations[19,20] due to the presence of massive chromatin extrusions, nuclear compression and bifurcation sites, and the emergence of chromatin segments that are not covered by any nuclear envelope marker tested.

**Loss of PPP1R12A compromises the nuclear compartment**. The discontinuity of nuclear envelope markers around DNA segments suggested that the nuclear compartment might be compromised in cells lacking PPP1R12A or PPP1CB. Consistent with this hypothesis, cells depleted of PPP1R12A showed a threefold increase in the fraction of cells with cytoplasmic pro-myelocytic leukemia protein (PML) aggregates, structures of 0.1 to 1 μm in diameter that normally reside in the nucleus and are too large to pass through nuclear pores (Fig. 1c)[21]. To directly interrogate the integrity of the nuclear compartment, we

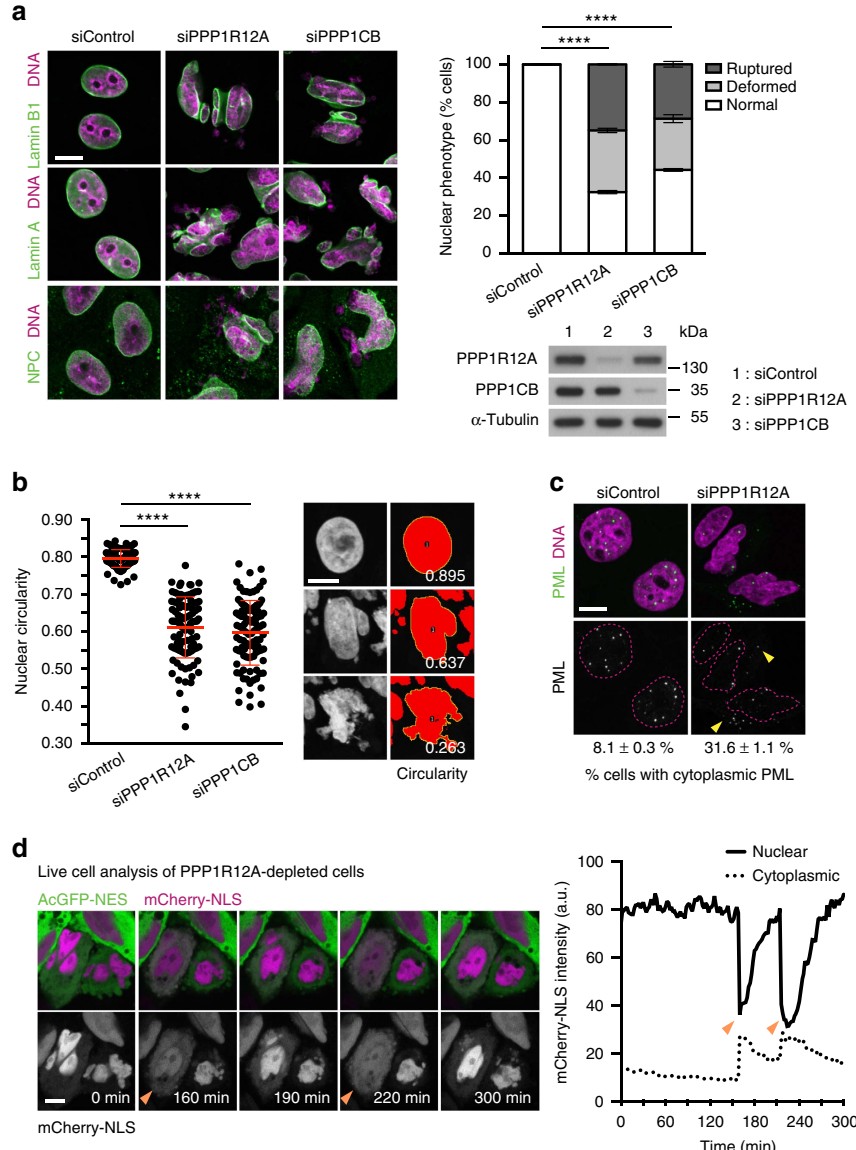

**Figure 1 | Depletion of PPP1R12A and PPP1CB causes nuclear deformation and nuclear envelope rupture. (a–c)** HeLa Kyoto cells transfected with PPP1R12A or PPP1CB siRNA were fixed 56 h after transfection. (**a**) Cells were stained with the indicated nuclear envelope markers and DAPI (left panel). Quantification of phenotype (top right panel). Immunoblot analysis of siRNA-transfected cells (bottom right panel). Abnormal nuclear shape (deformed) and nuclear envelope discontinuity (ruptured) are quantified. Error bars, s.d. of three independent experiments ($n > 200$ cells each). (**b**) PPP1R12A and PPP1CB siRNA-transfected cells were stained with DAPI and nuclear circularity was calculated for each nucleus (left panel). Red bars show mean and s.d. of three experiments. Representative nuclear shapes and their circularity indices are shown (right panel). Error bars, s.d. of three independent experiments ($n > 100$ nuclei each). (**c**) Localization of PML in PPP1R12A-depleted cells was analysed by immunofluorescence staining. The filled arrowheads indicate cytoplasmic PML signals. Percentages obtained from three independent experiments ($n > 200$ cells each). Mean of three experiments ± s.e.m. is shown. (**d**) Time-lapse microscopy of PPP1R12A-siRNA-transfected HeLa Kyoto cells transiently expressing AcGFP-NES and mCherry-NLS (left panel). Recording was started 40 h after transfection (0 min). mCherry-NLS signal intensity in the nucleus and cytoplasm was quantified over time (right panel). The filled arrowheads indicate nuclear rupture events characterized by the sudden efflux of mCherry-NLS from the nucleus into the cytoplasm. Graphs in **a,b** show ****$P < 0.0001$ (**a** $\chi^2$ test, **b** one-way ANOVA Tukey's multiple comparison test). Scale bars, 10 µm.

established a nuclear envelope barrier assay by co-expression of GFP fused to a nuclear export signal (AcGFP-NES) to mark the cytoplasm and mCherry fused to a nuclear localization signal (mCherry-NLS) to mark the nucleoplasm. In control interphase cells, the cytoplasmic and nuclear fluorescent marker proteins remained in their respective compartments (Supplementary Movies 6) (no exchange event recorded in 83 cells over 4 h). In contrast, in cells depleted of PPP1R12A, we detected nuclear envelope rupture events characterized by the sudden efflux of mCherry-NLS into the cytoplasm coinciding with the influx of AcGFP-NES into the nucleus (Fig. 1d; Supplementary Movies 7) (56 exchange events recorded in 292 cells over 4 h; average duration 15 min). During these events the intensity of mCherry-NLS in the nucleus and cytoplasm approached similar levels indicating free diffusion across the two compartments (Fig. 1d). We conclude that the loss of myosin phosphatase function causes rupture of the nuclear envelope, which leads to the emergence of DNA segments uncovered by nuclear envelope markers and compromises the integrity of the nuclear compartment.

**Unrestrained contractility causes nuclear envelope rupture**. Next, we aimed at elucidating the molecular basis of PPP1R12A–PPP1CB's function in safeguarding nuclear morphology and nuclear envelope integrity. Depletion of PPP1R12A did not strongly alter the cellular protein levels of lamin A and lamin B1 (Supplementary Fig. 4c,d). Depletion of PPP1R12A did also not induce apoptosis as judged by cytochrome *c* release from mitochondria and poly(ADP-ribose) polymerase cleavage (Supplementary Fig. 5a,b). PPP1R12A-depleted cells were less migratory than control counterparts indicating that disruption of nuclear morphology is unlikely to be driven by cell migration-associated processes (Supplementary Fig. 5c). Furthermore, fixed and live cell analysis revealed that the appearance of DNA outside the nuclear envelope was not the result of aberrant chromosome partitioning during mitosis or of defective nuclear envelope reformation during mitotic exit in PPP1R12A-depleted cells (Supplementary Fig. 6a–c; Supplementary Movies 8 and 9). Nuclei rather appeared to fragment shortly after the completion of mitosis in the absence of PPP1R12A (Supplementary Movies 9). PPP1R12A–PPP1CB has been reported to control the activity of the mitotic kinase PLK1 by dephosphorylating PLK1's T-loop at T210 (ref. 22). However, we did not observe a measurable increase in phosphoT210 PLK1 signal in HeLa cells depleted of either PPP1R12A or PPP1CB (Supplementary Fig. 6d). These observations suggest that alterations in the levels of lamin A and B1 proteins, the induction of apoptosis, cell migration and mitosis-related aberrations are not the culprits responsible for the striking nuclear integrity defects observed upon the loss of PPP1R12A–PPP1CB phosphatase.

The highly dynamic movement of nuclei and nuclear envelope indentations observed in cells depleted of PPP1R12A indicated the possible involvement of cytoskeletal elements and forces (Supplementary Movies 2,7 and 9). PPP1R12A–PPP1CB is known to antagonize cellular actomyosin contractility by dephosphorylating myosin regulatory light chain MYL9 (MRLC)[23]. MRLC is activated by phosphorylation of T18 and S19 at the hands of ROCK kinases downstream of the GTPase RhoA[9]. This raised the possibility that unrestrained contractility of actomyosin could be responsible for the nuclear damage in cells lacking PPP1R12A–PPP1CB phosphatase. Consistent with this hypothesis, depletion of PPP1R12A lead to increased MRLC phosphorylation (Fig. 2a; Supplementary Fig. 7a). Strikingly, treatment with the myosin ATPase inhibitor blebbistatin[24], the ROCK inhibitor Y-27632 (ref. 25) and the RhoA inhibitor C3 toxin[26] restored normal nuclear morphology, nuclear circularity and the integrity of the nuclear envelope in cells depleted of PPP1R12A and PPP1CB (Fig. 2b,c). In contrast, inhibition of myosin light chain kinase (MYLK) by addition of ML-7 had no effect (Fig. 2b). PPP1R12A-depleted cells treated with blebbistatin, Y-27632 and C3 toxin remained attached to the substratum demonstrating that the rescue of nuclear integrity was not caused by excessive cell rounding or detachment (Supplementary Fig. 7b). Co-depletion of ROCK1 and ROCK2 by RNAi or overexpression of a non-phosphorylatable version of MRLC, MRLC TASA (T18A S19A), also potently rescued the nuclear defects caused by the loss of myosin phosphatase (Fig. 3a; Supplementary Fig. 7c,d). Conversely, expression of a phospho-mimetic version of MRLC, MRLC TDSD (T18D S19D), was sufficient to induce nuclear fragmentation and nuclear envelope rupture in otherwise unperturbed cells (Fig. 3b; Supplementary Fig. 7d). Removal of the ROCK inhibitor Y-27632 from PPP1R12A-depleted cells in interphase caused nuclear fragmentation without passage through mitosis (Supplementary Fig. 8). This suggests that nuclear damage is not linked to defects in nuclear envelope reassembly during mitotic exit but can be triggered throughout interphase. We conclude that the nuclear

fragmentation and nuclear envelope rupture observed in PPPR12A–PPP1CB-depleted cells is not caused by a structural defect of the nuclear envelope but by damage inflicted by unrestrained actomyosin contractility in interphase cells. Our analyses suggest that MRLC phosphorylation by ROCK is the key target of PPP12A–PPP1CB phosphatase in safeguarding nuclear integrity, and that increased levels of MRLC phosphorylation are sufficient to drive nuclear dysmorphia.

**Mechanistic insights into actomyosin-driven nuclear damage.** How does the actomyosin network deform and damage the nucleus in cells lacking the PPPR12A–PPP1CB complex? We excluded a major role of linker proteins connecting F-actin and the nuclear envelope (SUN1, SUN2 and SYNE2)[4] in transmitting disruptive effects from the F-actin network to the nucleus (Supplementary Fig. 9). Interestingly, both the expression of active MRLC TDSD and PPP1R12A depletion led to the formation of actomyosin bundles in close proximity to the nucleus that bisected the main body of the nucleus and the smaller extruded segments (Fig. 3b,c; Supplementary Fig. 10a–c). Correlative light and electron microscopy and three-dimensional reconstruction of myosin phosphatase-depleted cells revealed the presence of actomyosin bundles on top of nuclear compression regions proximal to nuclear envelope rupture sites (Fig. 3c; Supplementary Fig. 10b,c). More than 80% of nuclear compression and nuclear envelope rupture sites were associated with F-actin in PPP1R12A-depleted cells (Supplementary Fig. 10c). This suggests that cytoplasmic actomyosin bundles could generate contractile forces that may squeeze the nucleus to such an extent that the nuclear lamina ruptures. Consistent with this hypothesis, genetic inactivation of lamin A in mouse embryonic fibroblasts (*LMNA* − / −  MEFs)[19] rendered nuclear integrity exquisitely sensitive to the effects of PPP1R12A depletion (Fig. 4a; Supplementary Fig. 11). Thus, the rigidity provided by lamin A protects the nuclear envelope from the impact of actomyosin activity and possibly forces.

Enhancing actomyosin contractility through PPP1R12A or PPP1CB depletion led to nuclear rupture in HeLa, MDA-MB-231, A549, SiHa and HT1080 cancer cells but had little effect on nuclear integrity in telomerase-immortalized hTERT RPE-1 cells and BJ-5ta primary fibroblasts (Supplementary Fig. 12). Comparison of primary vulval squamous cell carcinoma (VSCC) cells and vulval carcinoma-associated fibroblasts (VCAF) isolated from the same patient confirmed that depletion of PPP1R12A had a stronger effect in cancer cells (Fig. 4b). These observations suggest that transformed cells may be more amenable to force-mediated deformation of the nucleus than normal cells, possibly due to reduced nuclear stiffness in the former[27].

**Actomyosin is a determinant of nuclear shape and dynamics.** We observed that the nuclei of *LMNA* − / −  cells were less circular than controls even in under basal conditions and that nuclear circularity in these cells could be restored by addition of blebbistatin (Fig. 4a). This suggested that actomyosin contractility could be a major determinant of nuclear shape and drive nuclear dysmorphia in cancer. Consistent with this hypothesis, suppressing actomyosin activity by treatment with blebbistatin significantly increased nuclear circularity in cancer cell lines of different tissue origins, including the breast cancer cell line MDA-MB-231 (Fig. 5a). These data indicate that the nuclear atypia characteristic of many tumour cells is, at least in part, driven by actomyosin activity.

To extend our observations beyond cell culture to an *in vivo* setting, we generated MDA-MB-231 cells stably expressing H2B-mCherry and AcGFP-LAP2β. Depletion of PPP1R12A in MDA-

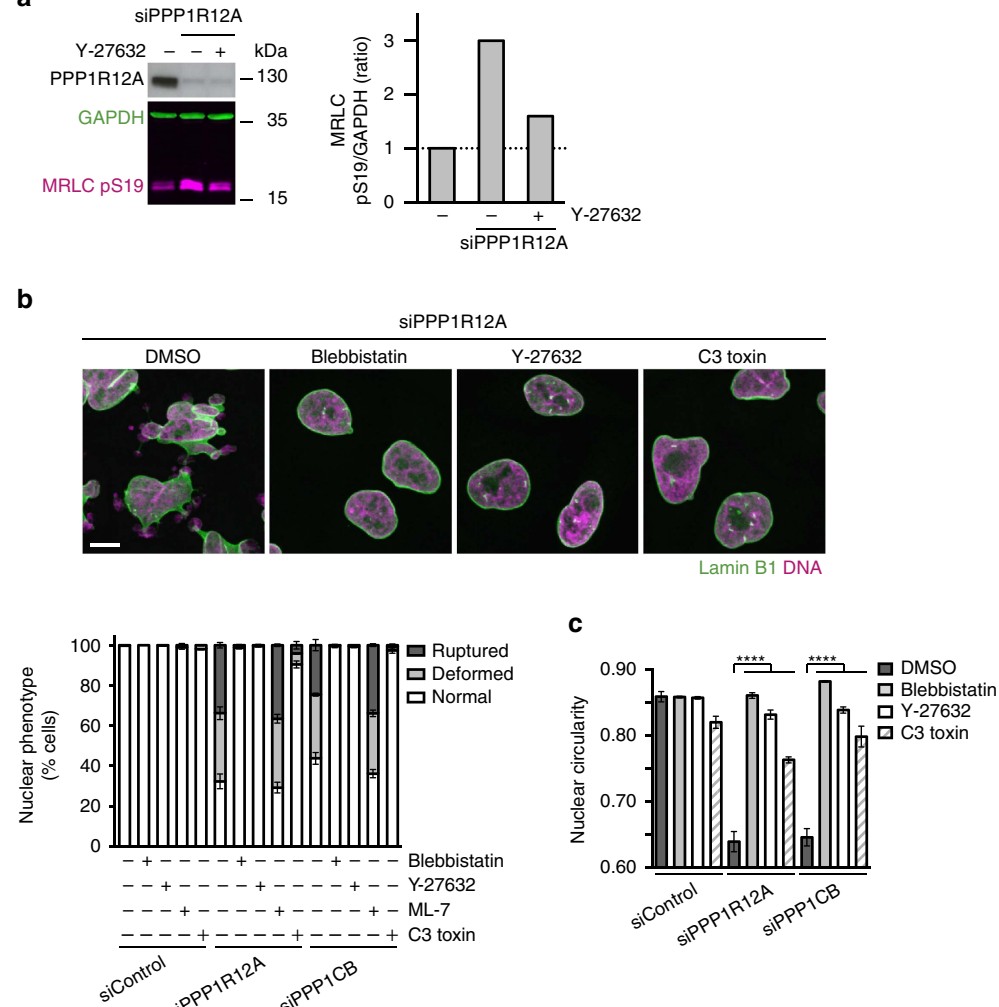

**Figure 2 | Actomyosin contractility drives nuclear deformation and rupture in PPP1R12A and PPP1CB-depleted cells.** (**a**) HeLa Kyoto cells were harvested 56 h after transfection with the indicated siRNA duplexes and analysed by quantitative immunoblotting. (**b,c**) HeLa Kyoto cells transfected with non-targeting control, PPP1R12A or PPP1CB siRNA were fixed 56 h after transfection and processed for immunofluorescence analysis. (**b**) siRNA-transfected cells were treated with 5 μM blebbistatin, 5 μM Y-27632, 10 μM ML-7, or 0.4 μg ml$^{-1}$ C3 toxin for 24 h before fixation. Representative immunofluorescence images are shown in the top panel. Abnormal nuclear shape (deformed) and nuclear envelope discontinuity (ruptured) are quantified in the bottom panel. Scale bar, 10 μm. (**c**) Quantification of nuclear circularity using the data from images acquired in **b**. Error bars, s.d. of three independent experiments ($n > 200$ cells each) (**b,c**). Graph shows ****$P < 0.0001$ (one-way ANOVA Tukey's multiple comparison test).

MB-231 cells cultured *in vitro* resulted in enhanced nuclear deformation and nuclear envelope rupture that were suppressed upon treatment with the ROCK inhibitor Y-27632 (Supplementary Fig. 13a). MDA-MB-231 cells stably expressing H2B-mCherry and AcGFP-LAP2β were injected under the nipple of nude mice. Once tumours had formed, intravital imaging was used to monitor dynamic changes in nuclear morphology (Fig. 5b,c). Nuclear morphologies were observed with a similar circularity score *in vivo* (Supplementary Fig. 13b) as *in vitro* (Fig. 5a). Importantly, regions of chromatin devoid of nuclear envelope were observed to form and be resolved in some cells *in vivo* (Fig. 5c; Supplementary Movie 10 and Supplementary Fig. 14). To determine if actomyosin contractility was driving the dynamic nuclear morphology, we treated tumours with the ROCK inhibitor Y-27632 (Supplementary Movies 11 and 12). The dynamic deformation of nuclei in xenografts was quantified by measuring parameters of individual nuclei over time and then calculating the variance of the values recorded for each nucleus (Fig. 5d). A high variance indicates dynamically changing nuclear

morphology. ROCK inhibition reduced the variance of nuclear perimeter, circularity and aspect ratio (Fig. 5d). In addition, ROCK inhibition increased nuclear circularity and reduced the nuclear perimeter in xenografts (Supplementary Fig. 13b). In line with our *in vitro* observations, these data strongly suggest that actomyosin contractility drives dynamic nuclear deformation and possibly nuclear envelope rupture *in vivo*. Further, dynamic nuclear deformation was not restricted to migratory cells, which is consistent with our observation that PPP1R12A depletion reduces migration *in vitro* (Supplementary Fig. 5c). Indeed, the large number of non-migratory cancer cells, compared to migratory cancer cells, meant that more dynamic nuclear deformation events were observed in non-migratory cells in imaged tumours (Supplementary Fig. 13c).

**Unrestrained actomyosin activity drives genome instability.** Next, we investigated whether nuclear deformation and rupture caused by actomyosin contractility was linked to genome

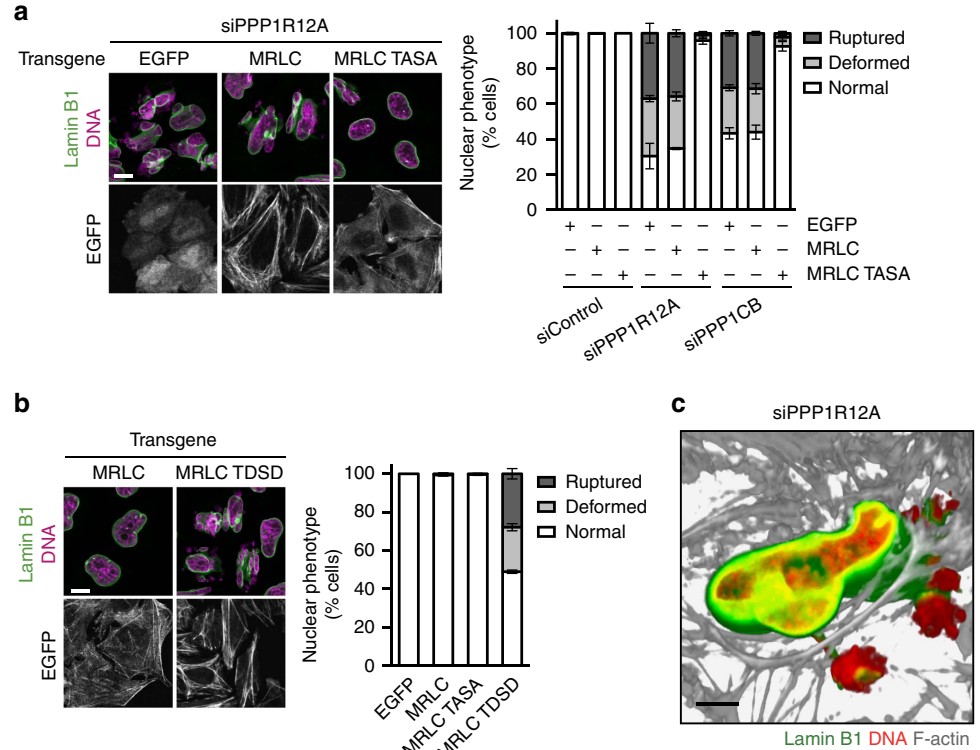

**Figure 3 | Deregulation of MRLC phosphorylation causes ectopic actomyosin fibre formation and nuclear deformation.** (**a**) HeLa Kyoto cells were transduced with lentiviral particles encoding EGFP, MRLC-EGFP wildtype or MRCL-EGFP T18A S19A (TASA). 15 h after infection, cells were transfected with the indicated siRNA duplexes and processed for immunofluorescence analysis 56 h after transfection. Only EGFP signal-positive cells were scored. Representative cell images (left panel) and quantification of nuclear phenotype (right panel) are shown. (**b**) HeLa Kyoto cells were infected with lentiviral particles encoding MRLC-EGFP wildtype or a T18D S19D mutant (TDSD) and processed for immunofluorescence analysis 56 h after infection. Only EGFP signal-positive cells were scored. Representative cell images (left panel) and quantification of nuclear phenotype (right panel) are shown. (**c**) PPP1R12A-depleted cells were fixed and stained for with anti-lamin B1 antibodies, DAPI and fluorophore-conjugated phalloidin. Cells were scanned in 0.1 μm sections using confocal laser scanning microscopy. The 3D image was reconstituted and rendered using Imaris software. Error bars, s.d. of three independent experiments (n > 200 cells each) (**a,b**). Scale bars, 10 μm.

instability. Depletion of PPP1R12A resulted in an increase in the signal intensity of the DNA damage markers γ-H2AX and 53BP1 in interphase nuclei (Fig. 6a,b). The magnitude of the γ-H2AX increase observed in PPP1R12A-depleted cells was similar to the one caused by treatment with the DNA polymerase inhibitor aphidicolin (Fig. 6a). Importantly, nuclear γ-H2AX signals in PPP1R12A-depleted cells were restored to close to baseline levels by addition of the ROCK inhibitor Y-27632 (Fig. 6a). Nucleotide analogue labelling experiments demonstrated that γ-H2AX induction occurred with a similar incidence in both replicating and non-replicating cells after depletion of PPP1R12A (Supplementary Fig. 15). This suggests that the DNA damage inflicted by unrestrained actomyosin contractility is not restricted to S phase but may occur throughout interphase. While DNA damage marked by γ-H2AX foci was distributed throughout the nucleus in most PPP1R12A-depleted cells (Fig. 6a), we observed striking examples in which the γ-H2AX signal was highly enriched and locally concentrated in chromatin segments that were protruding from the main nucleus and that lacked nuclear envelope markers (Fig. 6c). This suggests that nuclear deformation and nuclear envelope rupture could result in localized DNA damage consistent with previous observations in micronuclei that arose through mitotic segregation errors[28,29].

Finally, to probe the consequences of the DNA damage observed in interphase, we used mitotic chromosome spreading. To allow mitotic entry of cells with damaged chromosomes, siRNA-transfected cells were treated with the DNA damage checkpoint inhibitor caffeine. Mitotic chromosome spreads revealed that depletion of PPP1R12A led to an increased frequency of gross structural chromosome aberration, such as sister chromatid breaks and hypocondensation of chromosome segments (Fig. 7). In extreme cases, the majority of chromosomes of a karyotype were shattered in cells lacking PPP1R12A. Inhibition of ROCK strongly suppressed the emergence of chromosomal aberrations in cells lacking PPP1R12A (Fig. 7). Remarkably, these analyses describe a genome instability phenotype that is dependent on actomyosin contractility. Our data suggest that unrestrained actomyosin activity causes physical stress upon the nucleus driving nuclear deformation, nuclear envelope rupture and genome instability.

## Discussion

Altered nuclear morphology is a recurring feature of tumour histology, yet the molecular mechanisms that perturb nuclear shape are not well understood. Perturbation of the nuclear lamina has been reported to lead to altered nuclear morphology in several studies[30,31]. The shape variation and increased deformability of leukocyte nuclei correlates with lower lamin expression[32]. However, nuclear morphology is not just dependent on the physical properties of the nuclear envelope, but is also critically influenced by the forces exerted on the nucleus. We demonstrate that actomyosin contractility is a major determinant of nuclear shape and integrity in cell culture and *in vivo*.

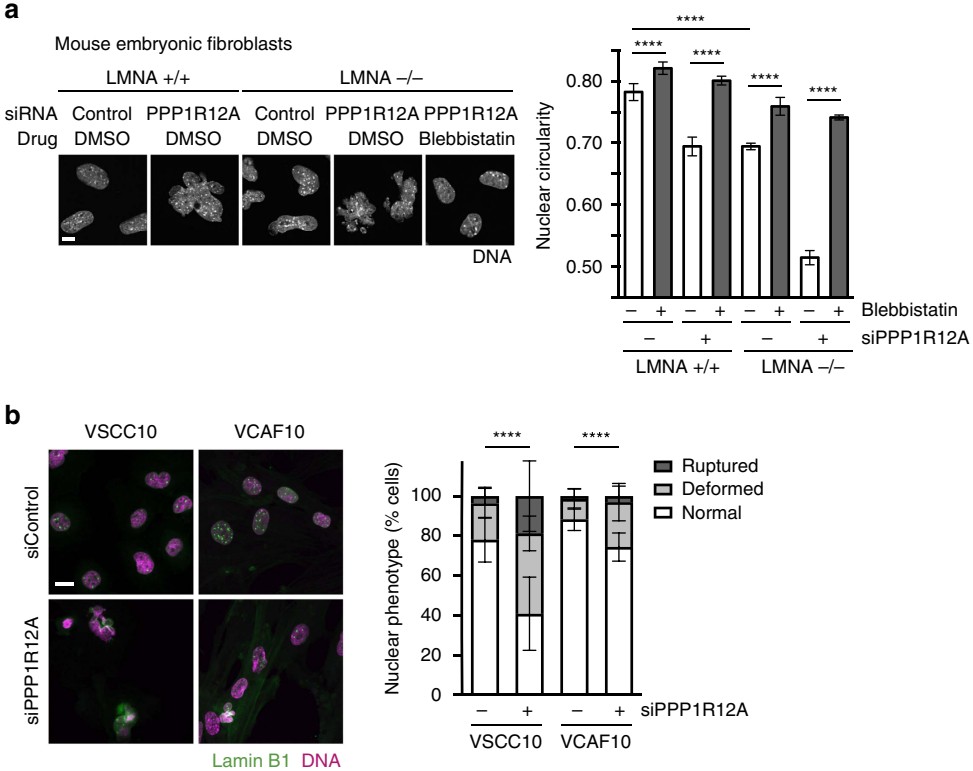

**Figure 4 | Impact of PPP1R12A depletion on nuclear morphology in _LMNA_ − / − and patient-derived cells.** (**a**) _LMNA_ + / + and − / − mouse embryonic fibroblasts were transfected with the indicated siRNA duplexes and stained with DAPI 56 h after transfection. Cells were exposed to DMSO solvent control or 5 µM blebbistatin for 24 h before fixation. Representative images (left panel) and quantification of nuclear circularity (right panel) are shown. Error bars, s.d. of three independent experiments ($n > 100$ nuclei each). Scale bar, 10 µm. (**b**) Vulval squamous cell carcinoma cells (VSCC10) and vulval cancer-associated fibroblasts (VCAF10) primary cells were transfected with the indicated siRNAs for 48 h. Representative cell images (left panel) and quantification of nuclear phenotype (right panel) are shown (VSCC10 siControl; $n = 100$, siPPP1R12A; $n = 82$, VCAF10 siControl; $n = 89$, siPPP1R12A; $n = 78$ cells per condition). Scale bar, 20 µm. Graphs show ****$P < 0.0001$ (**a** unpaired $t$ test; **b** $\chi^2$ test).

Inhibition of actomyosin contractility increases nuclear circularity in a panel of cancer cell lines. If deregulated by loss of the myosin phosphatase PPP1R12A–PPP1CB, unrestrained actomyosin activity and possibly actomyosin-dependent forces have detrimental effects on nuclear integrity including nuclear fragmentation, nuclear envelope rupture, loss of the nuclear compartment barrier and genome instability. The corollary of this conclusion is that myosin phosphatase prevents the cell nucleus and the genome from being attacked by actomyosin contractility. Our results suggest that thick cytoplasmic actomyosin fibres that are formed upon loss of myosin phosphatase locally compress the nucleus leading to nuclear envelope rupture and chromatin protrusions. This mechanism does not require protein complexes that link cytoplasmic cytoskeletal filaments to the nuclear envelope (such as LINC). Our data are consistent with the notion that the stiffness of the nuclear envelope conferred by LMNA moderates the impact of actomyosin activity on nuclear morphology and integrity. Unrestrained actomyosin activity had more pronounced effects on nuclear integrity in cancer cells than in telomerase-immortalized or primary cells. This observation could be explained by the emerging view that the nuclei of malignant cells are softer than that of normal cells[27]. Our work identifies the RhoA–ROCK axis, which is frequently deregulated in tumours[33], as the pathway driving nuclear damage through promoting the phosphorylation of myosin regulatory light chain. A related actomyosin-dependent mechanism might be utilized during apoptotic nuclear disintegration when ROCK cleavage at the hands of caspases causes enhanced contractility[34]. In the future, it will be important to interrogate whether oncogenic driver pathways that impact on Rho family GTPases alter nuclear shape and genome stability via the actomyosin cytoskeleton.

Actomyosin fibres and contractility influence both cell shape and nuclear shape[35,36]. Since our study does not assess cell shape directly, we cannot formerly exclude the possibility that deregulating actomyosin activity in myosin phosphatase-depleted cells or contractility inhibitor-treated cells causes nuclear shape changes indirectly via influencing cell shape. However, three observations strongly support the conclusion that actomyosin fibres directly impact the shape and integrity of nuclei in cells lacking myosin phosphatase in our study. First, most nuclear deformation and rupture sites are associated with actomyosin fibres in cells lacking myosin phosphatase. Second, myosin phosphatase-depleted cells remain attached to the substratum when nuclear integrity is restored upon treatment with contractility regulators. Lastly, it is difficult explain the dramatic nuclear deformation and multiple nuclear envelope rupture phenotypes observed in myosin phosphatase-depleted cells by invoking changes in cell shape only.

Physical forces and actomyosin structures have an impact on nuclear morphology and integrity. Forces acting on the nucleus of cells migrating through small gaps can lead to compression and disruption of the nuclear envelope as well as to DNA damage[37,38]. A perinuclear actin system stimulated by the formin protein FMN2 was recently shown to protect nuclear integrity during extravasation and metastasis[39]. Actin can also have disruptive effects on the cell nucleus. The combined inactivation of two

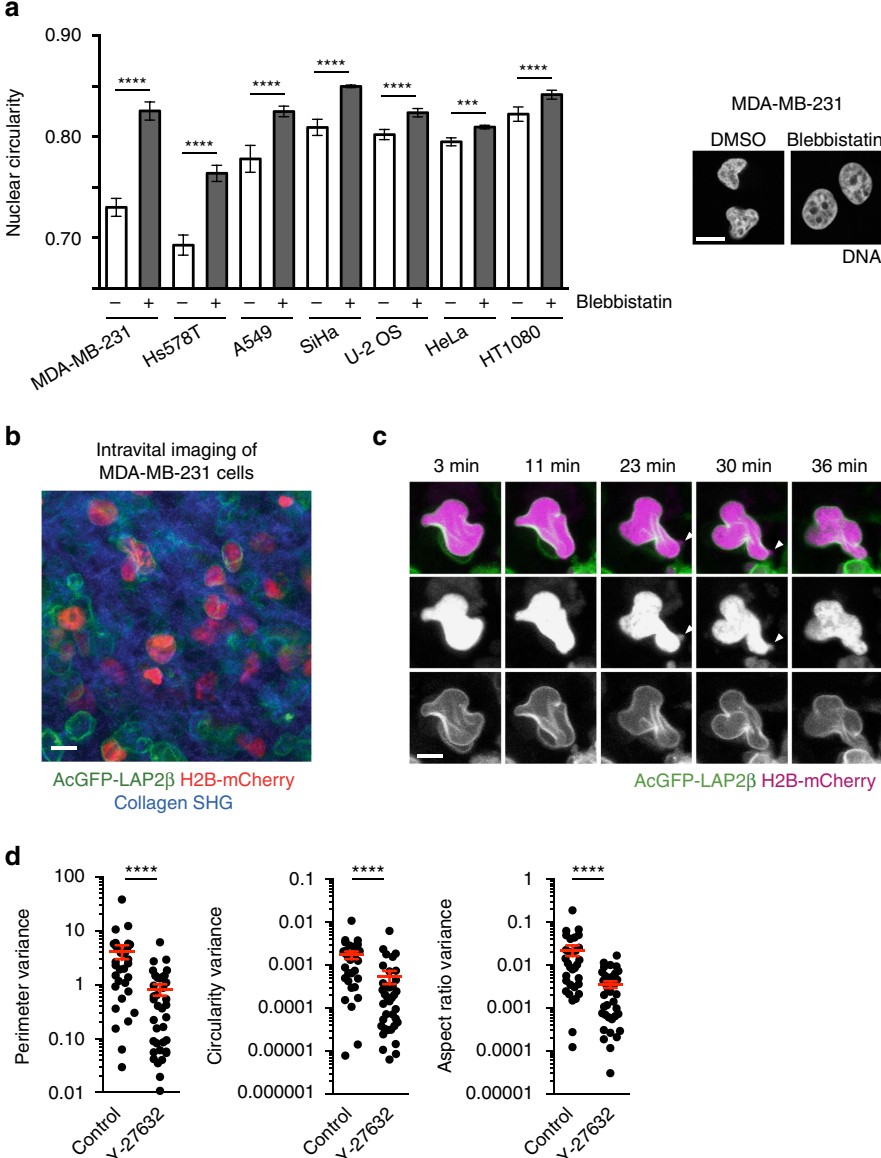

**Figure 5 | Actomyosin contractility is a determinant of nuclear shape and dynamics. (a)** Cancer cell lines derived from different tissues were treated with 5 µM blebbistatin for 24 h, fixed and stained with DAPI. Quantification of nuclear circularity (left panel) and representative images of the breast cancer cells line MDA-MB-231 (right panel) are shown. Error bars, s.d. of three independent experiments ($n > 100$ nuclei each). Scale bar, 10 µm. **(b)** Intravital imaging of an MDA-MB-231 xenograft tumour expressing AcGFP-LAP2β and H2B-mCherry. Collagen is shown in blue. Scale bar, 20 µm. **(c)** Intravital time-lapse series of an MDA-MB-231 tumour expressing AcGFP-LAP2β and H2B-mCherry. The arrowheads indicate a potential nuclear envelope rupture event. Images represent fields of 30 µm × 30 µm. Scale bar, 10 µm. **(d)** Quantification of nuclear perimeter, circularity and aspect ratio variance over time from intravital MDA-MB-231 imaging movies **(c)**. Variance values for individual cells of control and Y-27632-treated tumours are plotted (control; $n = 32$, Y-27632; $n = 37$ cells each). Graph shows ***$P = 0.0002$, ****$P < 0.0001$ (unpaired $t$ test).

F-actin severing proteins has been shown to promote the ectopic formation of stress fibres that cause nuclear deformation in epithelial cells[40]. Recent findings in cultured human cells also suggest that actin-based nucleus confinement in conjunction with LINC complexes can promote nuclear envelope rupture[41]. Our work on PPP1R12A–PPP1CB and the aforementioned studies suggests that nuclear deformation can occur in an actomyosin force-dependent manner that is not coupled to cell migration. Indeed, the absolute number of dynamic nuclear deformation events observed in non-motile cells was higher than in motile cells in the imaged tumour. The mechanism we describe is likely to be relevant for the control of nuclear shape and genome integrity in tumours even before the onset of invasion.

One of the striking findings emerging from our work is that unrestrained actomyosin activity can promote genome instability. PPP1R12A-depleted cells show increased γ-H2AX signals and 53BP1 foci, indicative for underlying DNA damage such as DNA double strand breaks. Importantly, the increase in γ-H2AX levels observed in PPP1R12A-depleted cells was dependent on actomyosin contractility. On the basis of nucleotide analogue incorporation experiments, the DNA damage inflicted by actomyosin contractility was not restricted replicating cells but likely occurs throughout interphase. The fact that the DNA damage signals were not aligned with actomyosin fibres bisecting the nucleus in most cells suggests that the impact of physical forces may not directly cause DNA damage. Instead, the transient

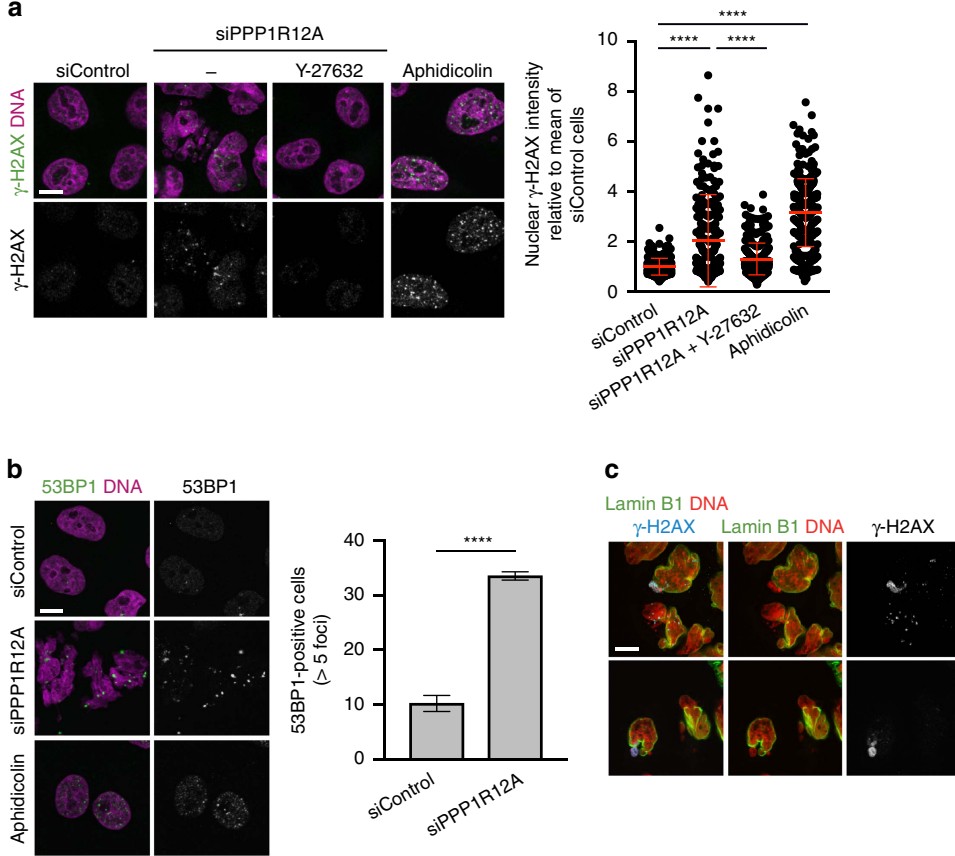

**Figure 6 | Induction of DNA damage markers by unrestrained actomyosin activity. (a,b)** HeLa Kyoto cells transfected with non-targeting control (siControl) or PPP1R12A-siRNA duplexes were processed for immunofluorescence staining 56 h after transfection. **(a)** Cells were treated with 5 μM Y-27632 or 400 nM aphidicolin for 24 h before fixation. Cells were stained with an anti-γ-H2AX antibody and DAPI. Representative images (left panel) and the quantification of γ-H2AX nuclear signal intensity (right panel) are shown. Mean nuclear intensities of individual cells are plotted. **(b)** Cells were stained with an anti-53BP1 antibody and DAPI. Representative images (left panel) and quantification of 53BP1-positive cells (right panel) are shown. Cells with more than five 53BP1 foci were scored as 53BP1-positive cells. **(c)** HeLa cells transfected with siPPP1R12A were stained with anti-lamin B1, γ-H2AX and DAPI. Error bars, s.d. of three independent experiments (n > 100 nuclei each) (**a** and **b**). Graph shows ****$P < 0.0001$ (**a** one-way ANOVA Tukey's multiple comparison test; **b** unpaired $t$ test). Scale bars, 10 μm.

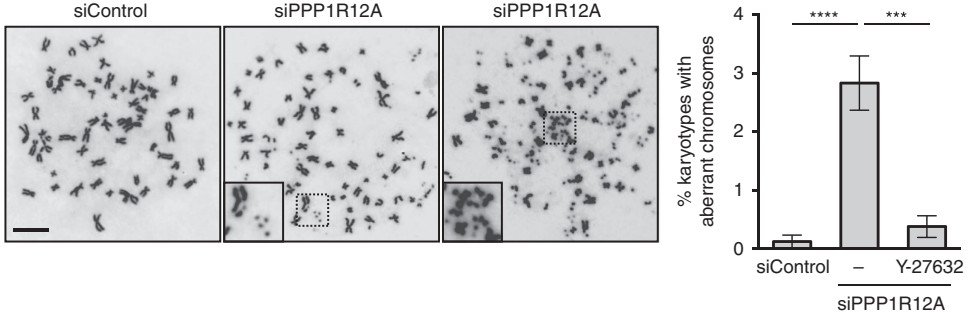

**Figure 7 | Loss of PPP1R12A causes contractility-dependent gross chromosomal aberrations.** Cells were treated with 5 μM Y-27632 for 24 h before fixation. Mitotic cells were collected by shake-off after the treatment with 330 nM nocodazole and 2 mM caffeine for 5 h. Mitotic chromosomes were analysed by chromosome spreading and Giemsa staining. Example karyotypes (left panel) and quantification of karyotypes with abnormal chromosomes (right panel) are shown. Error bars, s.d. of three independent experiments (n > 300 spreads each). Graph shows ***$P = 0.0001$, ****$P < 0.0001$ (one-way ANOVA Tukey's multiple comparison test). Scale bars, 10 μm.

loss of the nuclear compartment resulting in the efflux and dilution of DNA repair and metabolism factors may be driving genome instability. This hypothesis is supported by studies showing that nuclear envelope collapse is linked to genome instability and chromothripsis in micronuclei[28,29]. Whether actomyosin contractility-inflicted nuclear fragmentation and nuclear envelope rupture can result in the shattering and erroneous reassembly of one or a few chromosomes that is

characteristic of chromothripsis[42] remains to be tested. The chromosomal aberrations observed in PPP1R12A-depleted cells provide support for this hypothesis. Furthermore, although most extruded chromatin segments remain connected to the main nuclear body in PPP1R12A-depleted cells, the highly localized γ-H2AX signal observed in extruded chromatin regions lacking nuclear envelope markers in a subset of cells are reminiscent of the situation observed in micronuclei[29]. The essential DNA damage kinase ATR was recently found to relocalize to nuclear membranes upon mechanical stress where it could modulate nuclear envelope plasticity and chromatin association to the nuclear envelope[29]. It will be interesting to interrogate the subcellular distribution of ATR as well as the consequences of selective ATR inhibition in cells depleted of myosin phosphatase.

We found that PPP1R12A depletion leads to frequent and rapid loss of the nuclear compartment involving efflux bursts of soluble nuclear proteins into the cytoplasm. The transient nature of these events (average duration 15 min) suggests that emerging gaps in the nuclear envelope are resealed quickly and efficiently. Using correlative light and electron microscopy, we observed that chromatin regions that are not covered by the nuclear envelope marker LAP2ß are surrounded by a membranous structure (Supplementary Fig. 10b). Co-labelling experiments with the translocation pore marker SEC61B suggest that these membranes are derived from the rough endoplasmatic reticulum or from a section of the nuclear membrane that was left behind following rupture the nuclear lamina (Supplementary Fig. 10d). This raises the possibility that the rapid resealing of nuclear leaks may involve the association of chromatin with rough endoplasmatic reticulum membranes prior to the re-establishment of a nuclear lamina. Myosin phosphatase-depleted cells have the potential to serve as a useful model for studying the mechanisms underlying the sealing and repair of nuclear envelope rupture events in the future. Recent reports implicated the ESCRT-III membrane fission complex in sealing nuclear envelope rupture events upon nuclear confinement[37,38]. Whether ESCRT-III is also involved in repairing actomyosin-inflicted nuclear envelope rupture events remains to be tested.

To conclude, we demonstrate that nuclear dysmorphia of cancer cells and nuclear shape in general are controlled by the balance of opposing ROCK and PPP1R12A–PPP1CB phosphatase activities on myosin regulatory light chain. Hyper-activated and phosphorylated myosin light chain can generate sufficient actomyosin contractility and possibly forces to deform, and even break the nuclear envelope triggering genome instability.

## Methods

**Phosphatase siRNA screen.** A phosphatase siRNA library targeting 264 genes was obtained in 96 well format with pools of four individual siGENOME siRNA duplexes per well from Dharmacon (see Supplementary Data 1 for list of genes). HeLa Kyoto cells were transfected with siRNA at 30 nM using DharmaFECT 1 (Dharmacon). Cells were cultured for 72 h and fixed in –20 °C methanol. Fixed cells were stained using an α-tubulin antibody and 4′,6-diamidino-2-phenylindole (DAPI). Images were acquired on a Zeiss Axio Imager M2 microscope using a Plan-Neofluar × 40/1.3 numerical aperture (NA) oil objective lens (Zeiss) equipped with an ORCA-ER camera (Hamamatsu) and controlled by Volocity 6.0.1 software (Perkin Elmer). Images were subsequently scored for the fraction of nuclei with abnormal morphology ($n > 100$ cells).

**Reagents.** Target, supplier and catalogue number of the siRNA pools used for the phosphatase siRNA screen are listed in Supplementary Data 1. Antibodies, other siRNA duplexes and chemicals/reagents used throughout this study are listed in Supplementary Tables 1, 2 and 3, respectively. Details including suppliers, catalogue numbers and antibody dilutions are listed in the three tables.

**Antibodies.** Primary antibodies used for immunoblotting and immuno-fluorescence microscopy were rabbit anti-PPP1R12A (Santa Cruz Biotechnology sc-25618; immunoblotting 1:500), mouse anti-PPP1CB (Abcam ab53315; immunoblotting 1:50,000), mouse anti-lamin A (Abcam ab8980; immunofluorescence 1:1,000), mouse anti-lamin A (Cell Signaling Technology 4777; immunoblotting 1:2,000), rabbit anti-lamin B1 (Abcam ab16048; immunoblotting 1:5,000, immunofluorescence 1:1,000), mouse anti-LAP2ß (BD Biosciences 611000; immunofluorescence 1:1,000), mouse anti-NPC (Abcam ab50008; immunofluorescence 1:500), rabbit anti-SUN1 (Novus Biologicals NBP1-87396, immunofluorescence 1:400), rabbit anti-SUN2 (Abcam ab124916, immunofluorescence 1:100), mouse anti-SYNE2 (Thermo Scientific K20-478-5, immunofluorescence 1:1,000), mouse anti-PML (Abcam ab96051, immunofluorescence 1:500), mouse anti-ROCK1 (BD Biosciences 611137; immunoblotting 1:250), rabbit anti-ROCK2 (Santa Cruz Biotechnology sc-5561; immunoblotting 1:1,000), mouse anti-PLK1 (Santa Cruz Biotechnology sc-17783; immunoblotting 1:2,000), rabbit anti-PLK1 pT210 (Cell Signaling Tech 5472; immunoblotting 1:1,000), mouse anti-cyclin B (Santa Cruz Biotechnology sc-245; immunoblotting 1:2,000), rabbit anti-MRLC pS19 (Cell Signaling Tech 3671; immunoblotting 1:50, immunofluorescence 1:1,000), mouse anti-aurora B (BD Biosciences 611082; immunofluorescence 1:500), mouse anti-γ-H2AX (Millipore 05-636; immunofluorescence 1:600), mouse anti-cytochrome c (Life Technology 33-8200; immunofluorescence 1:25), mouse anti-GFP (Roche 11814460001; immunoblotting 1:1,000, immunofluorescence 1:1,000), mouse anti-GAPDH (Abcam ab8245; immunoblotting 1:40,000), mouse anti-AcGFP (Clontech 632381; immunofluorescence 1:1,000), rabbit anti-53BP1 (Novus Biologicals NB305-305; immunofluorescence 1:1,200), rabbit anti-cleaved PARP (Cell Signalling Tech 5625; immunofluorescence 1:400), mouse anti-α-tubulin (Sigma T6024; immunoblotting 1:20,000), rabbit anti-ß-tubulin (Cell Signalling Tech 5346; immunoblotting 1:2,000), phalloidin Alexa Fluor 488 (Life Technology A12379; immunofluorescence 1:500) and phalloidin Alexa Fluor 564 (Life Technology A12380; immunofluorescence 1:500).

HRP-conjugated secondary antibodies (GE Healthcare) were used to detect proteins on immunoblotting using chemiluminescence. For quantitative fluorescence immunoblotting, secondary antibodies conjugated to IRDye 680 and IRDye 800 (LI-COR; 1:5,000) were used for detection. Cross-adsorbed secondary antibodies conjugated to Alexa 488, Alexa 594, Alexa 568 or Alexa 647 (Life technology; 1:500) were used for immunofluorescence detection. Antibody information is also listed in Supplementary Table 1.

**Plasmids.** PPP1R12A (NM_002480) and PPP1CB (NM_002709) were inserted into pIRESpuro3-FLAc (FL denotes FLAG epitope; Ac denotes Aequorea coerulescens green fluorescent protein 1 (AcGFP))[43]. To create siRNA-resistant alleles, the PPP1R12A sequence TTCCATTTCT was changed to CAGCATCAGC, and the PPP1CB sequence TGGTGTTTC was changed to CGGCGTGAG by introducing synonymous mutations. The resulting plasmids are named pIRESpuro3-PPP1R12Ar-FLAc and pIRESpuro3-PPP1CBr-FLAc, respectively. Rat LAP2β (amino acids 244-453) was inserted into pIRESpuro3-AcFL[44] to obtain pIRESpuro3-AcFL-LAP2β. SEC61B (NM_006808.2) was inserted into pIRESpuro3-AcFL to obtain pIRESpuro3-AcFL-SEC61B. pIRESpuro3-AcGFP-NES was constructed by fusing the sequence encoding the nuclear export signal (NES) of HIV-1 Rev (amino acids PLQLPPLQRLTLN) to AcGFP. pIRESneo3-mCherry-NLS was constructed by fusing the sequence encoding the nuclear localization signal (NLS) of SV40 large T antigen (amino acids GGPKKKRKV) to mCherry. pLVXpuro-MRLC-EGFP WT, TASA (T18A S19A) and TDSD (T18D S19D) were constructed by inserting rat MRLC (MYL12B NP_059039.2)-EGFP with or without the indicated point mutations into pLVXpuro (Takara, Clontech).

**Cell culture and transfection.** Cell lines used in this study were cultured in DMEM supplemented with 10% fetal bovine serum. HeLa Kyoto cells were obtained from the Research Institute of Molecular Pathology (Vienna, Austria). The cell lines A549, DLD-1, HT1080, MCF-7 and MDA-MB-231 were obtained from HPA Cultures (formerly known as ECACC). The cell lines BJ-5ta, hTERT-RPE1, SiHa and U-2 OS were obtained from ATCC. HeLa Kyoto, MDA-MB-231, A549, SiHa, U-2 OS, MCF-7, DLD-1, hTERT RPE-1 and BJ-5ta cell lines were verified by STR profiling. All cell lines were tested negatively for mycoplasma contamination. Plasmid transfection was performed using FuGENE 6 (Roche). For siRNA transfection, cells were cultured for 15 h or more before transfection. siRNA duplexes (listed in Supplementary Table 2) were transfected at 30 nM using DharmaFECT 1. Culture medium was changed 24 h after transfection. Cells were usually cultured for 56 h after siRNA transfection before performing analyses. Immortalized LMNA +/+ and −/− mouse embryonic fibroblasts were kindly provided by Colin Stewart[19]. Vulval squamous cell carcinoma (VSCC10) and vulval carcinoma-associated fibroblast (VCAF10) were isolated from a vulval squamous cell carcinoma patient at Barts Hospital and the London NHS Trust following Research Ethics Committee approval 15/EE/0151. Informed patient consent was obtained.

**Establishment of stable cell lines.** To establish HeLa Kyoto cells stably expressing PPP1R12A-FLAc and PPP1CB-FLAc, cells were transfected with pIRESpuro3-PPP1R12Ar-FLAc and pIRESpuro3-PPP1CBr-FLAc, respectively, and selected in

medium supplemented with 0.3 µg ml$^{-1}$ puromycin (Sigma). HeLa Kyoto cells stably expressing AcFL-LAP2β were established by transfection with pIRESpuro3-AcFL-LAP2β followed by selection in medium supplemented with 0.3 µg ml$^{-1}$ puromycin. HeLa Kyoto H2B-mCherry cells[44] were transfected with pIRESpuro3-AcFL-LAP2β and selected in medium supplemented with 0.3 µg ml$^{-1}$ puromycin and 400 µg ml$^{-1}$ G418 (GIBCO) to establish HeLa Kyoto cells stably expressing AcGFP-LAP2β and H2B-mCherry. To establish MDA-MB-231 cells stably expressing AcGFP-LAP2β and H2B-mCherry, MDA-MB-231 cells were first transfected with pIRESpuro3-AcFL-LAP2β and selected in medium supplemented with 0.25 µg ml$^{-1}$ puromycin. AcGFP-positive cells were then used for transfection with pIRESneo3-H2B-mCherry and selected in medium supplemented with 400 µg ml$^{-1}$ G418.

**Lentivirus production and infection.** HEK293FT cells were co-transfected with pLVXpuro-MRLC-EGFP (WT, TASA or TDSD) or pLVXpuro-EGFP and the 2nd generation packaging system plasmids psPAX2 (Addgene, 12260) and pMD2.G (Addgene, 12259) using Lipofectamine 2000 (Invitrogen). Viral particles were collected at 24 and 48 h after transfection. HeLa Kyoto cells were infected with lentiviral particles in the presence of 8 µg ml$^{-1}$ polybrene (Sigma).

**Immunoblotting.** Cells were lysed in Laemmli buffer. Extracts were incubated at 95 °C for 5 min followed by brief sonication. The protein concentration in extracts was measured using the Bradford protein assay reagent (Bio-Rad). Samples were subjected to SDS-PAGE, transferred to polyvinylidene difluoride membranes. Membranes were probed with primary antibodies and subsequently with secondary antibodies conjugated to horseradish peroxidase. Protein bands were detected by using ECL system (GE Healthcare). For quantitative immunoblotting, secondary antibodies conjugated to IRDye 800CW and IRDye 680RD (LI-COR) were used. Protein band intensity was analysed using Odyssey (LI-COR). Full-sized immunoblots are shown in Supplementary Fig. 16.

**Immunofluorescence microscopy.** HeLa Kyoto cells were fixed with 4% formaldehyde at room temperature (RT) for 20 min or in methanol at −20 °C overnight and processed for immunofluorescence microscopy (IF) as described[45,46]. For 53BP1 staining, cells were treated with pre-extraction buffer (10 mM Pipes, pH 7, 100 mM NaCl, 300 mM sucrose, 1.5 mM MgCl$_2$, 5 mM EDTA and 0.5% Triton X-100) for 5 min on ice, then fixed with 4% formaldehyde for 20 min at RT. Primary antibodies used are listed in Supplementary Table 1. Alexa fluorophore-conjugated secondary antibodies were used for detection. DNA was stained using DAPI. Images were acquired using an Olympus FV1000D (InvertedMicroscopeIX81) confocal laser scanning microscope equipped with a PlanApoN ×60/1.40 NA Oil Sc objective lens controlled by FV10-ASW software. For visual inspection and scoring a Zeiss Axio Imager M1 microscope using Plan Neofluor ×40/1.3 oil objective lens (Zeiss) equipped with an ORCA-ER camera (Hamamatsu) and controlled by Volocity 5.5.1 software (Improvision) was used.

**Statistical tests.** P values were calculated using the paired t test, one-way ANOVA Tukey's multiple comparison tests and the χ$^2$ test of Prism Graphpad software.

**Measurement of nuclear circularity.** Cells were fixed in with 4% formaldehyde and stained with DAPI. Images of nuclei were acquired using an Olympus FV1000D laser scanning confocal microscope. Acquired nuclear images were analysed using the circularity function of Fiji software (https://fiji.sc)[47] and employing the formula: circularity = 4π(area/perimeter$^2$).

**EdU labelling.** HeLa Kyoto cells were transfected with siRNA and cultured for 54 h. Cells were treated with 10 µM EdU (Click-iT Plus EdU Imaging Kit, Molecular Probes) for 2 h before fixation. Incorporated EdU was detected according to the instructions of the Click-iT Plus EdU Imaging Kit. After EdU detection, cells were processed for γ-H2AX and DAPI staining.

**Live cell imaging.** HeLa Kyoto cells stably expressing AcGFP-LAP2β and H2B-mCherry were transfected with siRNA and cultured for 50 h before recording. Prior to start of recording, the culture medium was changed to phenol-red free and CO$_2$ independent medium (Invitrogen). Images were acquired as 4 z-planes 3 µm apart every 3 min at 37 °C using a Nikon TE2000 microscope equipped with a Plan Fluor ×60/1.4 DIC H Lens (Nikon), a PerkinElmer ERS Spinning disk system, a Digital CCD C4742-80-12AG camera (Hamamatsu), and controlled by Volocity 6.0.1 software. Cell images were projected with z-stacks and analysed by ImageJ. For analysis of cell migration, MDA-MB-231 cells were transfected with either control or PPP1R12A-siRNA using Lipofectamine 2000 (Invitrogen). The following day the cells were trypsinized and seeded in CO$_2$ independent media (Invitrogen) in 24 well MatTek dishes (MatTek Corporation). Six to eight hours after seeding the cells were imaged for 20 h using phase contrast microscopy using an adapted Zeiss

LSM510 microscope. Cell tracking was performed using the Manual Tracking plugin in Fiji software.

**Nuclear envelope barrier assay.** Hela Kyoto cells were transiently transfected with pIRESneo3-mCherry-NLS and pIRESpuro3-AcGFP-NES using FuGENE6. After 6 h, non-targeting control or PPP1R12A-siRNA was transfected using DharmaFECT 1. Time-lapse movies were performed 40 h after siRNA transfection. Prior to the start of recording, the culture medium was changed to phenol-red free and CO$_2$ independent medium (Invitrogen). Images were acquired as 3 z-planes 2 µm apart every 2 min at 37 °C using a Nikon TE2000 microscope equipped with a Plan Fluor ×60/1.4 DIC H Lens (Nikon), a PerkinElmer ERS Spinning disk system, a Digital CCD C4742-80-12AG camera (Hamamatsu), and controlled by Volocity 6.0.1 software. In each experiment, the number of cells and rupture events were quantified. Nuclear envelope rupture events were defined as the sudden efflux of mCherry-NLS into the cytoplasm and the sudden influx of AcGFP-NES into the nucleus without mitotic entry. Duration of each event was calculated as the time from marker influx/efflux until the mCherry-NLS began to accumulate in the nucleus again. To quantify mCherry-NLS signals over time, mean mCherry-NLS pixel intensity was measured by averaging four small circular regions manually placed in the nucleus or in the cytoplasm of a cell using ImageJ (https://imagej.nih.gov/ij/).

**Nuclear envelope reformation assay.** Hela Kyoto cells stably expressing AcGFP-LAP2ß were transiently transfected with pIRESneo3-mCherry-NLS using X-tremeGENE9. After 6 h, non-targeting control or PPP1R12A-siRNA was transfected using DharmaFECT 1. 24 h after plasmid transfection, cells were treated with 2 mM thymidine for 20 h, released for 6 h, arrested in mitosis with 50 ng ml$^{-1}$ nocodazole for 4 h. Subsequently cells were arrested in metaphase by addition of 10 µM MG132 for 2 h. Cells were released into anaphase by washing with CO$_2$ independent medium (Invitrogen) and recorded. Images were acquired as 4 z-planes 4 µm apart every 5 min at 37 °C using an Olympus FV1000D with a PlanApoN ×60/1.40 NA Oil Sc objective lens controlled by FV10-ASW software. Cell images were projected with z-stacks and analysed by ImageJ. Timing of nuclear envelope reformation was analysed by ImageJ and defined by the accumulation of mCherry-NLS signal in a region surrounded by AcGFP-LAP2β.

**3D reconstruction of cell images.** HeLa Kyoto cells transfected with siRNA were grown in LabTek chambered coverglass (Nunc) and processed for immunofluorescence staining as described above. Cell images were acquired using an Olympus FV1000D with a PlanApoN ×60/1.40 NA Oil Sc objective lens controlled by FV10-ASW software. Z-stacks were acquired at 0.1 µm step size across the entire cell volume. 3D images were reconstructed and rendered using Imaris software ver.6.4 (Bitplane).

**Correlative light and electron microscopy.** HeLa Kyoto cells expressing AcFL-LAP2ß and H2B-mCherry were transfected with control or PPP1R12A-siRNA and cultured for 56 h. Cells were incubated with 2 µM SiR-actin for 2 h and then fixed in 4% PFA/2.5% glutaraldehyde in 0.1 M phosphate buffer (pH 7.4) for 15 min. After fluorescence imaging, coverslips were processed using the Ellisman NCMIR protocol; post-fixed in 2% osmium tetroxide/1.5% potassium ferrocyanide for 1 h, on ice, then incubated in 1% weight per volume thiocarbohydrazide for 20 min before a second staining with 2% osmium tetroxide, and then incubated overnight in 1% aqueous uranyl acetate at 4 °C. Following this, the cells were stained with Walton's lead aspartate for 30 min at 60 °C. Cells were dehydrated through an ethanol series on ice, and incubated in a 1:1 propylene oxide/Durcupan resin mixture. Finally, the coverslips were embedded in Durcupan resin according to the manufacturer's instructions (TAAB Laboratories Equipment) and sectioned (70 nm). Images were acquired using a Tecnai Spirit Biotwin (FEI Company, Eindhoven, Netherlands) transmission electron microscope. Light and electron microscopy overlays were generated using the TurboReg plugin of Fiji software, with an affine transformation. Final adjustment was done by eye in Illustrator CS5.1.

**Intravital imaging.** Intravital imaging was performed under permission granted in UK Home Office licence PPL70/8380 following approval by the London Research Institute ethical review panel. Briefly, one million MDA-MB-231 cells stably expressing AcGFP-LAP2β and H2B-mCherry were injected in 20% Matrigel (BD Bioscience) under the fourth nipple of ICRF Nu/Nu mice. Intravital imaging was performed when tumours were between 5 and 12 mm in diameter. Mice were anaesthetized using isofluorane in oxygen before a small incision was made to expose the tumour margin. The mouse with the exposed tumour was then placed on an inverted laser scanning microscopy (Zeiss LSM 780) and viewed through a coverglass with a rubber ring to help stabilise the tumour against the breathing of the mouse. A small volume of warm PBS (GIBCO) was placed in the ring to prevent the tumour from drying. The tumour was maintained at physiological temperature by a combination of a thermostatically controlled heated stage and blanket over the tumour. A Ti-Sapphire laser tuned between 820 and 850 nm was used for second harmonic imaging of collagen fibres. Single photon excitation was

performed on AcGFP and mCherry using 488 and 561 nm lasers. Images were acquired every 60 s (unless explicitly stated otherwise) for 30–40 min. Imaging was performed within 20–80 μm of the tumour margin. For Y-27362 treatment, the tumour was topically bathed in 50 μM Y-27632 within the rubber ring used for imaging. 30 min was allowed to enable diffusion of Y-27632 into the tumour margin. As before, imaging was performed within 20–80 μm of the tumour margin. Analysis was performed using Fiji software. The nuclear morphology of individual cells was traced at five intervals within the time-lapse data set (typically 7 or 8 min apart). At each time point the perimeter, sphericity and aspect ratio were measured. Both the average and the variance of these measurements at the five time points were calculated for each cell.

**γ-H2AX quantification.** HeLa cells transfected with siRNA duplexes or treated with aphidicolin were fixed with 4% PFA. Cells were stained using DAPI and γ-H2AX antibodies. The intensity of nuclear γ-H2AX for each cell was measured by Fiji software. Individual cell data were normalized to the mean signal intensity of cells transfected with control siRNA.

**Chromosome spreading.** To facilitate the analysis of structural chromosome aberrations, HeLa cells were incubated for 5 h in medium containing 330 nM nocodazole (Sigma) and 2 mM caffeine. Mitotic HeLa cells were swelled in a hypotonic solution (DMEM:deionized water at 1:2 ratio) for 6 min at RT. Subsequently, cells were fixed with Carnoy's buffer (freshly made) for 15 min at RT and spun down, this fixation step was repeated four times. The suspension of cells in Carnoy's buffer (100 μl) was dropped on a clean slide from about 80 cm distance and let dry at RT. Slides were incubated with 3% Giemsa in PBS for 6 min at RT. After drying, slides were mounted with DPX mountant (Sigma).

**Data availability.** The authors declare that the data supporting the findings of this study are available within the paper and its Supplementary Information files or from the authors upon request.

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

## Acknowledgements

We thank Ken Blight, Matthieu Piel and Colin Stewart for sharing reagents and providing technical expertise. Work in the laboratory of MP was supported by the EMBO Young Investigator Programme and Cancer Research UK. E.S. and M.M. were funded by

the Francis Crick Institute, which receives its core funding from Cancer Research UK (FC001144), the UK Medical Research Council (FC001144) and the Wellcome Trust (FC001144). M.M. also received funding from Marie Curie Actions—Intra-European Fellowships #625496. T.T. and S.J.B. were supported by the Francis Crick Institute, which receives its core funding from Cancer Research UK (FC0010048), the UK Medical Research Council (FC0010048) and the Wellcome Trust (FC0010048); a European Research Council (ERC) Advanced Investigator Grant (RecMitMei); and a Wellcome Trust Senior Investigator Grant.

## Author contributions

T.T., M.M., M.P.S., M.R., L.C., E.S. and M.P. performed the experiments and analyses. T.T., M.M., M.P.S., L.C., S.J.B., E.S. and M.P. designed the experiments. M.L.B., K.S., M.H. and S.J.B. provided expert advice, resources and reagents. T.T., E.S. and M.P. wrote the manuscript.

## Additional information

**Competing interests:** M.P. is an employee of Boehringer Ingelheim RCV and declares no competing interest in connection with this work. The remaining authors declare no competing financial interests.

