## [Peer Review File · Nature Communications]

Editorial Note: The figure on page 7 is reprinted by permission from Macmillan Publishers Ltd: Scientific Reports 3 Article Number 1449, copyright 2013

Editorial Note: The figure on page 18 (Figure R3A) is sourced from Integr Biol (Camb). 4(11): 1406-14, copyright 2012

Reviewers' comments:

Reviewer #1 (Remarks to the Author):

In this manuscript, Takaki et al. investigate how the actin cytoskeleton controls nuclear morphology and how this impacts on genomic integrity. It is not well understood how mechanical forces of cytoskeleton govern the shape of cell nuclei, and how misregulation of cytoskeletal contractility leads to aberrant nuclear morphogenesis that are typical for cancer cells. Hence, this study will be very interesting for cell biologists as well as cancer researchers.

The authors identified by RNAi screening for nuclear shape regulators two subunits of the myosin phosphatase complex and show that they are important to maintain a round nuclear morphology, a sealed nuclear envelope, and a stable genome. Through a series of elegant and very rigorously controlled loss-of-function and gain-of-function experiments, the authors demonstrate that excessive actomyosin contractility deforms nuclei and damages DNA, in vitro as well as in a mouse tumor model. They further show that nuclear envelope stiffness, provided by the lamina, protects nuclei against deformation by excessive contractility of the actomyosin cytoskeleton.

These findings are exciting and very well supported by the data. I am convinced that this study will be highly relevant for a broad readership and thus strongly recommend publication in Nature Communications.

Reviewer #2 (Remarks to the Author):

This manuscript by Petronczki and colleagues describes the control of actomyosin contractility as a critical modulator of nuclear morphology and genome integrity in cancer cells. Using a phosphatase-focused RNAi screen, the authors uncovered myosin phosphatase (PPP1R12A/PPP1CB) as a key factor in maintaining nuclear shape. The authors elegantly dissect the molecular pathway involved in this process as well as the impact of PPP1R12A/PPP1CB loss on nuclear integrity. The myosin regulatory light chain MRLC is identified as the responsible myosin phosphatase target, and inhibition of upstream Rho-ROCK signaling, phospho-mutant MRLC or myosin ATPase inhibition are able to revert the nuclear rupture phenotype. Finally, the authors demonstrate a role for Rho-ROCK signaling in altering nuclear shape in tumors in vivo and show that this pathway, when perturbed, may contribute to genomic instability. Together, this is a compelling study that significantly advances our understanding of nuclear dysmorphia and its potential adverse consequences in cancer. The following issues should, however, be clarified:

1) The cause for the observed increase in genomic instability is speculative at this point. A more thorough investigation of the types of DNA lesions (i.e. double-strand and/or single-strand breaks) as well as possible cell cycle dependence would significantly strengthen this aspect of the manuscript. Are lesions a result of replication or do they occur throughout interphase? The authors propose F-actin contraction as a cause for nuclear rupture, implying the generation of "ruptured" chromosomes and, hence, DSBs, which can be tested (e.g. using alkaline versus neutral comet assays).

2) On a related note, the authors should place their work in the context of recent findings from the Foiani lab, demonstrating that mechanical stress is necessary to activate ATR during S phase and thereby facilitate DNA replication.

3) The bursts of mCherry export (Fig. 1d) are striking and suggest that nuclear rupture(s) also occur in bursts. The authors should at a minimum comment on this observation and discuss possible causes.

4) Only ~ ½ of the tested cell lines appear to respond to myosin phosphatase depletion (Fig. S11a). No effect is seen in two hTERT-immortalized, non-transformed cell lines. Why do the authors think that is? And what is the effect in primary cells?

5) The authors should include at least one additional marker for apoptosis.

Reviewer #3 (Remarks to the Author):

Recent work on nucleus structure has shown that the nuclear envelope (NE) is much more dynamic than previously realized. Rupture and repair of the NE has been shown to occur in laminopathy and cancer cells in culture and migrating cancer and immune cells. NE rupture is speculated to contribute to genome instability in cancer cells by causing DNA damage. One initiator of NE structure defects leading to NE rupture is increased tension on the membrane from either mechanical nucleus compression or assembly of actin bundles. Here Takaki et al. identify a mechanism that drives accumulation of contractile actin bundles resulting in increased NE structure defects and NE rupture and reinforces the idea that the actin cytoskeleton is a major determinant of nucleus morphology and stability. Specifically, they identify a phosphatase complex (PPP1R12A/PPP1CB) that is required to maintain nucleus circularity and integrity by antagonizing Rho-ROCK kinase activation of a myosin complex protein. They show that loss of the phosphatase causes an increase in actin bundles in the cell that correspond with 1) increased lobulation of the nucleus, 2) blebbing of chromatin through disruptions in the nuclear lamina, and 3) rupture of the NE causing loss of nucleus compartmentalization. In addition, this paper shows that increased actomyosin activity is correlated with increased DNA damage. The authors connect these in vitro findings to in vivo tumor conditions by demonstrating that xenograft nuclei become rounder when treated with actomyosin inhibitors. However they do not convincingly show that the cells have defects in NE stability or DNA damage accumulation prior to drug treatment and thus it is unclear what the significance of this finding is. Overall this paper does an excellent job characterizing a new mechanism that inhibits actin accumulation to prevent NE instability, but some additional control experiments are required to solidify these findings.

Major Issues:

1. The phrases describing the nucleus phenotypes are used inconsistently throughout the paper and it is unclear what the authors think is really happening. For instance, in an early description they say they observe “DNA fragments outside the nuclear lamina” yet there is no experiment demonstrating that these DNA structures represent a broken piece of a chromosome that is unconnected to the rest of the genome. In fact at least one of their images (Fig. 2f) has a thinner DNA band connecting these extended regions to the rest of the nucleus. Thus it is unclear whether the phenotype is fragmentation of a single nucleus into multiple nuclei or a single nucleus with multiple lobes or blebs. From their images it looks like they are seeing the formation of multiple nucleus blebs, some at the sites of lamina holes, as previously described (e.g. Sullivan et al., JCB, 1999, de Noronha et al., Science, 2001, Shimi et al, Genes Dev, 2008). A more careful description of the nucleus morphology and single plane images of where the lobes connect to the main nucleus would help to clarify this issue as well as a discussion of how similar these structures are to those previously described.

In addition, the authors use different three phrases - nucleus rupture, NE rupture, and lamina rupture – to describe defects in lamina structure. This becomes confusing since it is not clear when they are talking about lamina defects versus rupture of the nuclear membrane. The authors need to define these terms and stick with a single phrase to describe each event.

Finally, using the phrase “lamina rupture” implies that new holes are forming in the nuclear lamina. Although this is likely, to distinguish the formation of new holes from the expansion of existing holes the authors need to expand their description of the nuclear lamina in untreated

cells, ideally using super-resolution microscopy.

2. The authors use cell lines overexpressing Lap2B to conclude that nuclear envelope reassembly at mitosis is normal in phosphatase depleted cells, that actin accumulates at chromatin necks, and to observe the in vivo dynamics of MDA-MB-231 cells. However, Lap2B is a structural component of the nuclear lamina that interacts with the lamins and chromatin regulatory proteins and is required for proper nuclear envelope assembly. Thus, the authors need to show additional controls demonstrating that Lap2B overexpression does not have additional effects on nucleus and lamina morphology, and on the kinetics of NE assembly.

3. The authors need an additional control to show that phosphatase disruption does not cause loss of lamin proteins from the NE, which can cause similar shape changes and increased NE rupture. This needs to be examined in the presence and absence of actomyosin inhibitors as well since actin structure has been shown to regulate lamin levels and phosphorylation status.

4. One of the major observations in the authors' in vivo xenograft experiments is that, in the presence of actin, they see un-laminated chromatin, indicative of larger NE structural defects and similar to what they see in culture. However the MDA-MB-231 cells in culture shows little evidence of lamina disruption (Fig. S11), and they show no evidence of lamina rupturing in the xenograft. In addition, it is not clear in the movie that the red spot interpreted as un-laminated chromatin is part of the same nucleus or same cell as the laminated chromatin. Thus, their evidence seems to show only that MDA-MB-231 nuclei are soft and responsive to weak actin forces in xenografts, rather than that actin is driving significant changes in NE structure that could cause NE rupture or DNA damage in MDA-MB-231 cells.

5. The discussion of the result that increased actin contractility causes genome instability due to increased DNA damage is a little confusing. The karyotype images were taken from cells where the DNA damage checkpoint was inhibited by caffeine. Thus these images do not necessarily represent the real amount of chromatin fragmentation that would occur during mitosis as a result of increased damage. This should be clarified in the text.

Minor issues

1. The student's T-test is not sufficient to determine the significance for data with more than two samples. Graphs like Fig. 1b should be analyzed using a one-way ANOVA to determine the significance for the entire family in conjunction with a multiple comparisons post-test. In graphs such as Fig. 2c and 4a, where the interaction of multiple conditions are being tested (siRNA + drug) the authors need to use a 2-way ANOVA plus a multiple comparisons post-test and always include the data for the ctl siRNA + drug condition (Fig. 4a). In addition, the authors should use a chi-square analysis to assess the significance of the nominal data presented in graphs such as Fig 1a.

2. The authors rule out the possibility that defects in NE assembly could contribute to altered NE structure in their experiments. However, what they show is that mitosis is not necessary for these alterations. To rule out that NE assembly is not contributing they would have to do additional analysis of the timing of NE assembly, the extent of chromosome decompaction during reassembly, and the timing of protein recruitment during NE assembly in cells depleted of the phosphatase complex.

Response to reviewers' comments

General

We were very pleased at the overall positive nature of the comments and we are grateful for the constructive criticism and suggestions provided by the referees. In particular, that 'these findings are exciting and very well supported by the data' (reviewer #1), the work is 'a compelling study that significantly advances our understanding of nuclear dysmorphia' (reviewer #2), and that we do 'an excellent job characterizing a new mechanism that inhibits actin accumulation to prevent NE instability' (reviewer #3). We also note that reviewers #2 and #3 make several pertinent suggestions to improve the manuscript. We have addressed the points raised by the reviewers by conducting a series of new experiments and analyses as well as by making changes to the text. We provide new evidence about the nature of DNA damage induced by actomyosin forces, we have performed analysis of primary patient cells – both tumour and stroma, and have performed additional control experiments (impact of nuclear envelope markers on phenotype, lamin level analysis etc). In addition, we have expanded the discussion section and put our work in better context of the literature.

Please see below for a point-by-point response to the reviewers' comments.

Response to Reviewer #1

In this manuscript, Takaki et al. investigate how the actin cytoskeleton controls nuclear morphology and how this impacts on genomic integrity. It is not well understood how mechanical forces of cytoskeleton govern the shape of cell nuclei, and how misregulation of cytoskeletal contractility leads to aberrant nuclear morphogenesis that are typical for cancer cells. Hence, this study will be very interesting for cell biologists as well as cancer researchers.

The authors identified by RNAi screening for nuclear shape regulators two subunits of the myosin phosphatase complex and show that they are important to maintain a round nuclear morphology, a sealed nuclear envelope, and a stable genome. Through a series of elegant and very rigorously controlled loss-of-function and gain-of-function experiments, the authors demonstrate that excessive actomyosin contractility deforms nuclei and damages DNA, in vitro as well as in a mouse tumor model. They further show that nuclear envelope stiffness, provided by the lamina, protects nuclei against deformation by excessive contractility of the actomyosin cytoskeleton.

These findings are exciting and very well supported by the data. I am convinced that this study will be highly relevant for a broad readership and thus strongly recommend publication in Nature Communications.

Response: We are delighted that the reviewer expresses such strong support for the work and does not raise any issues that need addressing.

Response to Reviewer #2

This manuscript by Petronczki and colleagues describes the control of actomyosin contractility as a critical modulator of nuclear morphology and genome integrity in cancer cells. Using a phosphatase-focused RNAi screen, the authors uncovered myosin phosphatase (PPP1R12A/PPP1CB) as a key factor in maintaining nuclear shape. The authors elegantly dissect the molecular pathway involved in this process as well as the impact of PPP1R12A/PPP1CB loss on nuclear integrity. The myosin regulatory light chain MRLC is identified as the responsible myosin phosphatase target, and inhibition of upstream Rho-ROCK signaling, phospho-mutant MRLC or myosin ATPase inhibition are able to revert the nuclear rupture phenotype. Finally, the authors demonstrate a role for Rho-ROCK signaling in altering nuclear shape in tumors in vivo and show that this pathway, when perturbed, may contribute to genomic instability. Together, this is a compelling study that significantly advances our understanding of nuclear dysmorphia and its potential adverse consequences in cancer. The following issues should, however, be clarified:

Response: We are very pleased that the reviewer finds the work compelling and a significant advance.

1) The cause for the observed increase in genomic instability is speculative at this point. A more thorough investigation of the types of DNA lesions (i.e. double-strand and/or single-strand breaks) as well as possible cell cycle dependence would significantly strengthen this aspect of the manuscript. Are lesions a result of replication or do they occur throughout interphase? The authors propose F-actin contraction as a cause for nuclear rupture, implying the generation of "ruptured" chromosomes and, hence, DSBs, which can be tested (e.g. using alkaline versus neutral comet assays).

Response: We agree that it is important to clarify the type of DNA damage in more detail. To this end, we have now performed staining for 53BP1, which is recruited to double strand breaks (**new Figure 6b**). This shows that PPP1R12A depletion leads to a clear increase in 53BP1 foci indicating double strand breaks. Comet assays were not possible because embedding in agar undermines the anchorage to a substrate required for the generation of large contractile actomyosin forces. Using nucleotide analog incorporation experiments we have addressed the relationship between DNA damage marker induction and DNA replication in PPP1R12A-depleted cells. Our findings suggest that DNA damage in PPP1R12A-depleted cells is not restricted to S phase but can occur throughout interphase (**new Supplementary Figure 14**). We also provide data indicating that DNA damage markers in PPP1R12A-depleted cells can be focused and concentrated in DNA segments that lack nuclear envelope markers (**new Supplementary Figure 6c**). This finding is discussed in the context of work from the Pellman lab describing genome instability in micronuclei.

2) On a related note, the authors should place their work in the context of recent findings from the Foiani lab, demonstrating that mechanical stress is necessary to activate ATR during S phase and thereby facilitate DNA replication.

Response: We thank the reviewer for this suggestion. We now refer to this study and put the implications of our work in the context of the findings of cited study in the discussion section of our manuscript (**page 13**).

3) The bursts of mCherry export (Fig. 1d) are striking and suggest that nuclear rupture(s) also occur in bursts. The authors should at a minimum comment on this observation and discuss possible causes.

Response: In the revised version of the manuscript we discuss this phenomenon of nuclear ruptures and the breakdown of the nuclear compartment in more detail (see discussion section). Furthermore, we provide new data suggesting that the transient nature of these bursts could be explained by the association of extruded chromatin segments with SEC61-positive ER membranes (**new Supplementary Figure 10b**).

4) Only ~ ½ of the tested cell lines appear to respond to myosin phosphatase depletion (Fig. S11a). No effect is seen in two hTERT-immortalized, non-transformed cell lines. Why do the authors think that is? And what is the effect in primary cells?

Response: We agree that this is an intriguing result. Following the reviewer's suggestion, we have now probed the effect of PPP1R12A depletion in primary patient cells. Specifically, we isolated both the transformed cancer cell compartment and non-transformed stromal fibroblasts from a squamous cell carcinoma patient. PPP1R12A depletion in the cancer cells (VSCC10) led to a clear increase in nuclear dysmorphia, whereas the effect in the stromal fibroblasts (VCAF10) was less pronounced (**new Figure 4b**). These new data exclude that the differences between cell lines are related to the primary vs established cell lines or the length of time in culture. We believe that the differences between cell lines relate to the intrinsic physical properties of the nucleus. Analysis by The Physical Sciences – Oncology Centers Network (<http://www.nature.com/articles/srep01449> Figure 2f(iii) – shown below for the benefit of the reviewer) shows that the nuclei of transformed breast MDA-MB-231 cells are much softer when deformed by $>0.2\mu\text{m}$ than non-transformed MCF10A. This supports the emerging view that the nuclei of malignant cells are softer than that of normal cells. Consistent with this, we observe the greatest nuclear deformation in HeLa, MDA-MB-231, A549, and VSCC10 (all cancer cells) and almost no PPP1R12A loss-mediated damage of nuclear morphology in non-transformed hTERT-RPE1 cell and primary BJ fibroblasts. This point is now included in the discussion section of our revised manuscript.

Figure reproduced from Scientific Reports showing the different Young's modulus of the nucleus of MDA-MB-231 and MCF10A cells. Note that for deformation $>0.1\mu\text{m}$ the MDAMB231 cell nucleus is softer.

5) The authors should include at least on additional marker for apoptosis.

Response: We now show cleaved PARP as an apoptosis marker in addition to mitochondrial cytochrome c release (**new Supplementary Figure 5b**). This new dataset supports our conclusion that PPP1R12A depletion does not cause nuclear morphology changes through induction of apoptosis.

Response to Reviewer #3

Recent work on nucleus structure has shown that the nuclear envelope (NE) is much more dynamic than previously realized. Rupture and repair of the NE has been shown to occur in laminopathy and cancer cells in culture and migrating cancer and immune cells. NE rupture is speculated to contribute to genome instability in cancer cells by causing DNA damage. One initiator of NE structure defects leading to NE rupture is increased tension on the membrane from either mechanical nucleus compression or assembly of actin bundles. Here Takaki et al. identify a mechanism that drives accumulation of contractile actin bundles resulting in increased NE structure defects and NE rupture and reinforces the idea that the actin cytoskeleton is a major determinant of nucleus morphology and stability. Specifically, they identify a phosphatase complex (PPP1R12A/PPP1CB) that is required to maintain nucleus circularity and integrity by antagonizing Rho-ROCK kinase activation of a myosin complex protein.

They show that loss of the phosphatase causes an increase in actin bundles in the cell that correspond with 1) increased lobulation of the nucleus, 2) blebbing of chromatin through disruptions in the nuclear lamina, and 3) rupture of the NE causing loss of nucleus compartmentalization. In addition, this paper shows that increased actomyosin activity is correlated with increased DNA damage. The authors connect these in vitro findings to in vivo tumor conditions by demonstrating that xenograft nuclei become rounder when treated with actomyosin inhibitors. However they do not convincingly show that the cells have defects in NE stability or DNA damage accumulation prior to drug treatment and thus it is unclear what the significance of this finding is. Overall this paper does an excellent job characterizing a new mechanism that inhibits actin accumulation to prevent NE

instability, but some additional control experiments are required to solidify these findings.

Response: We were pleased with the recognition that we have characterized a new mechanism that prevents NE instability. We are also grateful for the various technical comments that have prompted us to perform some further experimental controls and to clarify our language.

1. The phrases describing the nucleus phenotypes are used inconsistently throughout the paper and it is unclear what the authors think is really happening. For instance, in an early description they say they observe “DNA fragments outside the nuclear lamina” yet there is no experiment demonstrating that these DNA structures represent a broken piece of a chromosome that is unconnected to the rest of the genome. In fact at least one of their images (Fig. 2f) has a thinner DNA band connecting these extended regions to the rest of the nucleus. Thus it is unclear whether the phenotype is fragmentation of a single nucleus into multiple nuclei or a single nucleus with multiple lobes or blebs. From their images it looks like they are seeing the formation of multiple nucleus blebs, some at the sites of lamina holes, as previously described (e.g. Sullivan et al., JCB, 1999, de Noronha et al., Science, 2001, Shimi et al, Genes Dev, 2008). A more careful description of the nucleus morphology and single plane images of where the lobes connect to the main nucleus would help to clarify this issue as well as a discussion of how similar these structures are to those previously described.

Response: The reviewer is correct that there is some ambiguity about whether DNA observed outside the nuclear envelope remains connected to the main body of the nucleus. We have now performed more careful analysis of confocal z-sections to address this. **Supplementary Figure 3b** confirms the reviewer’s observation that thin DNA connections to the nucleus remain. However, in 18% of cases there was no DNA strand connecting the ‘fragment’ to the nucleus. We therefore propose that nuclear envelope rupture typically leads to the extrusion of DNA into the cytoplasm, but that the extruded segments normally remain linked to the bulk of the nucleus. Only in more extreme circumstances, possibly when the forces involved are particularly high, does the extruded DNA become truly fragmented or is an entire chromosome extruded. We have adapted the text accordingly and have related the phenotype observed in myosin phosphatase depleted cells to previously described nuclear envelope aberrations in the suggested references.

In addition, the authors use different three phrases - nucleus rupture, NE rupture, and lamina rupture – to describe defects in lamina structure. This becomes confusing since it is not clear when they are talking about lamina defects versus rupture of the nuclear membrane. The authors need to define these terms and stick with a single phrase to describe each event.

Response: We thank the reviewer for requesting that we standardize our terminology to improve the readability of the manuscript. Our data suggest persistent, frequent and widespread rupture of the nuclear lamina and transient breaches of the nuclear membrane. We now use the term NE rupture to cover both ongoing phenomena as this does not imply

particular defects in the lamina, nor does it imply complete destruction of the nucleus.

Finally, using the phrase “lamina rupture” implies that new holes are forming in the nuclear lamina. Although this is likely, to distinguish the formation of new holes from the expansion of existing holes the authors need to expand their description of the nuclear lamina in untreated cells, ideally using super-resolution microscopy.

Response: We agree with the reviewer regarding the possible implications of the term lamina rupture. As mentioned above, we now use the more neutral term of nuclear envelope rupture. It would be fascinating to observe the process with higher spatio-temporal resolution; however, to properly distinguish nuclear pores and lamina sub-structures a resolution in the tens of nm is required (a nuclear pore is ~120nm across) and this is only achievable by PALM/STORM type of super-resolution microscopy methods, but we know from previous studies that organelles within highly contractile cells are moving at speeds ~0.1µm/s (analysis conducted for Tozluoglu et al Nature Cell Biology 2013, although it was not included in the final manuscript). Unfortunately, PALM/STORM cannot operate at the necessary frame rates for such fast motion. Therefore, while dynamic super-resolution imaging of nuclear envelope sub-structures would be very interesting, it is not possible with current technology.

2. The authors use cell lines overexpressing Lap2B to conclude that nuclear envelope reassembly at mitosis is normal in phosphatase depleted cells, that actin accumulates at chromatin necks, and to observe the in vivo dynamics of MDA-MB-231 cells. However, Lap2B is a structural component of the nuclear lamina that interacts with the lamins and chromatin regulatory proteins and is required for proper nuclear envelope assembly. Thus, the authors need to show additional controls demonstrating that Lap2B overexpression does not have additional affects on nucleus and lamina morphology, and on the kinetics of NE assembly.

Response: We agree that this is a key control. We now show in **new Supplementary Figure 4a&b** that expression of the live cell markers AcGFP-LAP2 β and H2B-mCherry does neither change the baseline circularity of cells, nor does it alter the penetrance of nuclear morphology aberrations caused by depletion of PPP1R12A. This finding is in line with our observation that control cells stably expressing AcGFP-LAP2 β and H2B-mCherry do not show bursts of nuclear protein leakage (no exchange event recorded in 83 cells over 4 h; Supplementary Movie 8). Also we would like highlight the fact that most of our fixed cell analyses of nuclear envelope markers was performed in cells that do not overexpress nuclear envelope marker proteins (e.g. Fig. 1a and Supplementary Figure 3a,b).

3. The authors need an additional control to show that phosphatase disruption does not cause loss of lamin proteins from the NE, which can cause similar shape changes and increased NE rupture. This needs to be examined in the presence and absence of actomyosin inhibitors as well since actin structure has been shown to regulate lamin levels and phosphorylation status.

Response: We now show in **new Supplementary Figure 4c & d** that depletion of PPP1R12A does not have a major impact on the levels of lamin A or lamin B as analysed by both western blot and immuno-fluorescence microscopy.

4. One of the major observations in the authors' in vivo xenograft experiments is that, in the presence of actin, they see un-laminated chromatin, indicative of larger NE structural defects and similar to what they see in culture. However the MDA-MB-231 cells in culture shows little evidence of lamina disruption (Fig. S11), and they show no evidence of lamina rupturing in the xenograft. In addition, it is not clear in the movie that the red spot interpreted as un-laminated chromatin is part of the same nucleus or same cell as the laminated chromatin. Thus, their evidence seems to show only that MDA-MB-231 nuclei are soft and responsive to weak actin forces in xenografts, rather than that actin is driving significant changes in NE structure that could cause NE rupture or DNA damage in MDA-MB-231 cells.

Response: We now provide the individual confocal sections to unambiguously show that the un-laminated H2B signal is associated with the cell, and not an out of plain artefact (**new Supplementary Figure 13b**).

5. The discussion of the result that increased actin contractility causes genome instability due to increased DNA damage is a little confusing. The karyotype images were taken from cells where the DNA damage checkpoint was inhibited by caffeine. Thus these images do not necessarily represent the real amount of chromatin fragmentation that would occur during mitosis as a result of increased damage. This should be clarified in the text.

Response: We thank the reviewer for pointing this out and now clarify the use of caffeine in the text (**page 11**). Caffeine was used to allow cells with damaged chromosomes to enter to mitosis. We propose that PPP1R12A suppresses actomyosin-driven chromatin fragmentation in interphase and the purpose of the experiment is to reveal the aberrations in chromosome structure accumulated prior to mitosis.

Minor issues

1. The student's T-test is not sufficient to determine the significance for data with more than two samples. Graphs like Fig. 1b should be analyzed using a one-way ANOVA to determine the significance for the entire family in conjunction with a multiple comparisons post-test. In graphs such as Fig. 2c and 4a, where the interaction of multiple conditions are being tested (siRNA + drug) the authors need to use a 2-way ANOVA plus a multiple comparisons post-test and always include the data for the ctrl siRNA + drug condition (Fig. 4a). In addition, the authors should use a chi-square analysis to assess the significance of the nominal data presented in graphs such as Fig 1a.

Response: We thank the reviewer for pointing out more appropriate statistical tests and have now performed ANOVA analysis in the **Figures 1b, 2c, and 4a** in line with his/her suggestions.

2. The authors rule out the possibility that defects in NE assembly could contribute to altered NE structure in their experiments. However, what they show is that mitosis is not necessary for these alterations. To rule out that NE assembly is not contributing they would have to do additional analysis of the timing of NE assembly, the extent of chromosome decompaction during reassembly, and the timing of protein recruitment during NE assembly in cells depleted of the phosphatase complex.

Response: We have added new data (**new Supplementary Figure 6c**) in which we measure the length of time for nuclear envelope reformation following mitosis. This analysis shows that there is no change in nuclear envelope reformation during mitotic exit in PPP1R12A depleted cells.

Reviewers' comments:

Reviewer #2 (Remarks to the Author):

The authors have satisfactorily addressed my concerns.

Reviewer #3 (Remarks to the Author):

In the revised manuscript the authors have done an excellent job of addressing my concerns.

I do have a comment on one of the new sections, the discussion of supplemental figure 10b, which shows that nucleus fragments associate with Sec61B-containing membranes. The discussion is a little confusing - the authors don't make clear that Sec61B labels the nuclear membranes as well as the ER, and thus all the fragments that have mCherry-NLS should have a rim of Sec61B. There were also two papers (Denais et al and Raab et al, Science 2016) that described a role for the ESCRTIII complex in membrane resealing after nuclear envelope rupture that should be mentioned in this section.

Reviewer #4 (Remarks to the Author):

1. This paper primarily identifies PPP1R12A and PPP1CB as 'proteins safeguarding nuclear integrity'. However, this discovery does not appear to be novel, as stated in the text: "Previous screens have also identified PPP1R12A and PPP1CB as factors controlling nuclear morphology (references 10, 11). That the lamina confers stiffness on the nucleus toward cytoplasmic forces is also well known.
2. The screen for nuclear shape was based on visual inspection, which makes it difficult to predict if others can reproduce this work. This is also a problem in other experiments (not the screen alone).
3. Circularity is quantified in later assays as a measure of irregularity. But circularity is low for elongated nuclei which are 'smooth' where no irregularities are present in the nuclear shape. Circularity is not necessarily a measure of irregularity, which makes it difficult to interpret the data and any differences.
4. Data in Figure 3B from different cancer cell lines show that nuclear circularity is higher in these cell lines when myosin is inhibited with blebbistatin. However, PPP1R12A and PPP1CB levels in these cell lines appear comparable (at least from a visual inspection of Fig. S12), therefore it is not clear how this data should be interpreted in the light of the mechanism that the paper proposes.
5. Relevant to point 4, treatment of cells with inhibitors, especially inhibitors like blebbistatin, change cell shape which can change nuclear shape, for example by rounding cells up, or causing a fragmented cytoplasm. It is not clear that the observed effects are due to actomyosin activity or indirect effects on the cell shape itself, or some other mechanism. For example, another possibility is that upon inhibition of myosin, cells with irregular nuclei detach preferentially, enriching the population in normal looking nuclei.
6. Figure 2f is very difficult to interpret in terms of F-actin filaments causing nuclear deformations given the low resolution. It is claimed that Figure 2e shows how actomyosin fibers 'bisect the nucleus and cause nuclear fragments'. But in Figure 2e, at least a few examples can be seen where actin fibers are not bisecting or touching the nucleus. Are all deformations of the nucleus due to impinging actomyosin fibers? The paper does not put forth a convincing mechanism for how the nucleus is fragmenting upon knockdown of PPP1R12A and PPP1CB.

7. The *in vivo* measurements are unconvincing. Nuclear abnormalities are not clearly visible in the picture provided (Figure 3c). It is not clear that the decrease in variance of the measured parameters necessarily indicates a smoother nucleus upon treatment with Y27632.

8. Could the results in Figure 4 be interpreted in another way? Chromosomal spreads are done by colliding cells with a solid surface which forces the chromosomes apart. Knockdown of phosphatases makes the cytoplasm stiffer which might cause breaks in chromosomes.

9. A direct comparison of actomyosin forces generated by the different cancer cell lines, along with a comparison of mechanical properties of the nucleus would be required along with evidence that myosin inhibition dynamically causes the highly abnormal nucleus to relax back to its 'smooth' shape.

Response to Reviewers' comments

We would like to thank Reviewers #2 & #3 for their further comments, and to thank you Reviewer #4 for the criticism and comments on our work. We were very gratified that reviewer #2 stated that we had 'satisfactorily addressed [my] concerns', and that reviewer #3 felt that we 'have done an excellent job of addressing [my] concerns'. We note that he/she had one remaining point around SEC61B localization requiring clarification. Unfortunately, the comments of reviewer #4 were not forwarded to us in October last year and we were only made aware of these comments recently, several days after the resubmission of our revised manuscript at the end of March. Nonetheless, we address numerous points raised by referee #4 by new data incorporated into the revised version, or by clarification of the text. We hope that our comments and the data below can clarify and address the points raised by referee #4 in this exceptional situation. This is in addition to the four months of new experimental work that has resolved the concerns of reviewers #2 & #3.

Response to Reviewer #3

1. I do have a comment on one of the new sections, the discussion of supplemental figure 10b, which shows that nucleus fragments associate with Sec61B-containing membranes. The discussion is a little confusing - the authors don't make clear that Sec61B labels the nuclear membranes as well as the ER, and thus all the fragments that have mCherry-NLS should have a rim of Sec61B. There were also two papers (Denais et al and Raab et al, Science 2016) that described a role for the ESCRTIII complex in membrane resealing after nuclear envelope rupture that should be mentioned in this section.

We thank the reviewer for raising this point. We have now modified the text in the discussion section to clarify that SEC61B-positive membranes can be derived from the rough ER or the nuclear membrane. Furthermore, we have cited the two papers demonstrating that ESCRT-III promotes the sealing of nuclear envelope rupture events upon nuclear confinement (**page 14**). Whether ESCRT-III is implicated in repairing actomyosin-inflicted nuclear rupture events is an interesting question for future research and now also mentioned in the discussion.

Response to Reviewer #4

1. This paper primarily identifies PPP1R12A and PPP1CB as 'proteins safeguarding nuclear integrity'. However, this discovery does not appear to be novel, as stated in the text: "Previous screens have also identified PPP1R12A and PPP1CB as factors controlling nuclear morphology (references 10, 11). That the lamina confers stiffness on the nucleus toward cytoplasmic forces is also well known.

The reviewer is correct that the finding that depletion of PPP1R12A and PPP1CB causes changes in nuclear morphology detected by DAPI staining was made previously. This fact is appropriately cited in our manuscript (**middle of page 4**). However, the type of nuclear alteration and the mechanism underlying it was unknown. Our work addresses these points. We for the first time describe the type of nuclear morphology and integrity defects occurring in myosin phosphatase-depleted cells and, importantly, we discover the mechanism underlying these nuclear aberrations. We show that loss of myosin phosphatase inflicts

nuclear damage, including nuclear envelope rupture, loss of the nuclear compartment, double-stranded DNA breaks, and genome instability, via unrestrained actomyosin activity. We further use intravital imaging to demonstrate nuclear rupture events in non-migrating cells *in vivo*, and provide evidence that ROCK-mediated actomyosin contractility regulates this process.

2. The screen for nuclear shape was based on visual inspection, which makes it difficult to predict if others can reproduce this work. This is also a problem in other experiments (not the screen alone).

The original siRNA screen was performed using visual inspection. The full data set derived from the screen is provided in Supplementary Table 1. siRNA data in general require downstream validation using siRNA pool deconvolution and transgenic rescue experiments, as we have performed successfully for PPP1R12A and PPP1CB. Thus, the primary screen result list can be utilized by other researchers to shortlist additional phosphatase genes as candidates for factors controlling cell division or nuclear integrity. In addition to PPP1R12A and PPP1CB, the catalytic subunit isoforms of PPP2A that are known to be required to preserve sister chromatid cohesion until metaphase, are highly scoring hits in our screen supporting the ability of our approach to identify relevant genes.

We would respectfully disagree with the comment regarding the quality of analyses presented in the paper. Data for many key findings in the paper are collected using quantitative image analysis algorithms and are supported by multiple experiments and approaches. The quality of our analyses was also commended by other referees.

3. Circularity is quantified in later assays as a measure of irregularity. But circularity is low for elongated nuclei which are 'smooth' where no irregularities are present in the nuclear shape. Circularity is not necessarily a measure of irregularity, which makes it difficult to interpret the data and any differences.

This is correct and the reason why we use additional classifiers in many of the figures. The 'normal/deformed/ruptured' categories distinguish a 'smooth' long nucleus from a 'rounder, more irregular one'. In our manuscript, the changes in nuclear shape between control cells, myosin phosphatase-depleted cells and depleted cells treated with contractility inhibitors are striking, highly significant using circularity analysis and discernable by looking at the images. This differences in the circularity metric are primarily a reflection of perimeter of the nucleus becoming more 'tortuous' upon PPP1R12A depletion, not a change in elongation, which would be most accurately scored by an 'aspect ratio' metric. The circularity analysis presented in our paper is in strong agreement with the other visual qualifiers (e.g. ruptured nuclear envelope). Below we now provide an additional **Figure (Figure R1)** showing that treatment with the ROCK inhibitor restores nuclear shape and integrity in a panel of cancer cell lines depleted of PPP1R12A. Therefore, we believe that the use of a combination of classifiers including circularity is a suitable and fit-for-purpose approach to analyze nuclear phenotypes in this study.

Treatment with the ROCK inhibitor Y-27632 suppresses nuclear aberrations in PPP1R12-depleted cells

Figure R1: A panel of cell lines was transfected with the indicated siRNA duplexes for 56 hrs and treated with 7.5 uM Y-27632 for 24 hrs before fixation as indicated. Nuclear morphology was subsequently analyzed by Lamin B1 and DNA staining. ($n > 200$ cells each)

4. Data in Figure 3B from different cancer cell lines show that nuclear circularity is higher in these cell lines when myosin is inhibited with blebbistatin. However, PPP1R12A and PPP1CB levels in these cell lines appear comparable (at least from a visual inspection of Fig. S12), therefore it is not clear how this data should be interpreted in the light of the mechanism that the paper proposes.

The experiment in the original Figure 3b demonstrates that actomyosin activity makes an important contribution to nuclear shape in a panel of cell lines even if not deregulated by depletion of myosin phosphatase. However, as the reviewer correctly points out the protein levels of myosin phosphatase subunits do not significantly differ between cell lines whose nuclei are more deformed by depletion of the phosphatase or more circularized by inhibition of contractility. The reason for this is that PPP1R12A is regulated by phosphorylation, and intersects with Rho-ROCK-MLC2 pathway activity at multiple levels (Kawano et al JCB 1999, Feng et al JBC 1999, Wilkinson et al NCB 2005). The difference between the cell lines is likely to relate to differences in activity that are related to post-translational modifications in PPP1R12A and additional differences, such as the expression of positive contractility regulators (such as RhoA, ROCK kinases, and the key RhoGEF Ect2) and of lamins. Below we provide an immunoblotting analysis of contractility regulators in a panel of cancer cell lines (**Figure R2**). The cell lines that show the most pronounced effect of PPP1R12A depletion have consistently high levels of Ect2, RhoA, ROCK1, and ROCK2. The lack of effect of PPP1R12A depletion in MCF-7 and DLD-1 cells is, quite possibly, due to the low levels of RhoA expression. This would be consistent with the data using C3 toxin to block RhoA in **Figure 2b**. Given the likely multifactorial nature of the differences in cellular responses in complex non-isogenic models, such as cancer cell lines, we decided not to speculate on this point in our manuscript. In isogenic systems we show the absence of lamin A sensitizes to the impact of actomyosin contractility (**Figure 4a**).

Figure R2: A panel of cell lines was transfected with the indicated siRNA duplexes for 56 hrs and analyzed by immunofluorescence microscopy ($n > 200$ cells) (left panel). Protein extract of the indicated cell lines were analyzed by immunoblotting (right panel).

5. Relevant to point 4, treatment of cells with inhibitors, especially inhibitors like blebbistatin, change cell shape which can change nuclear shape, for example by rounding cells up, or causing a fragmented cytoplasm. It is not clear that the observed effects are due to actomyosin activity or indirect effects on the cell shape itself, or some other mechanism. For example, another possibility is that upon inhibition of myosin, cells with irregular nuclei detach preferentially, enriching the population in normal looking nuclei.

We did not observe cell loss due to detachment or massive cell rounding following treatment with blebbistatin or Y-27632. A low concentration of blebbistatin (5 μ M) that was insufficient to abrogate cytokinesis was able to restore nuclear architecture in PPP1R12A and PPP1CB-depleted cells. Cells treated with contractility inhibitors at the concentrations used in our work showed reduced actomyosin fibers but remained attached to the surface – **these new data are shown in Supplementary Figure 7b**. Importantly, the Y-27632 washout experiment shown in **Supplementary Figure 8** of the revised manuscript uses live cell imaging to show the alleviation of ROCK inhibition (by washout of a ROCK inhibitor) causes nuclear morphology changes and nuclear envelope rupture in PPP1R12A-depleted cells. A corollary of this result is the conclusion that the suppression of nuclear damage by contractility inhibitors cannot be explained by cell loss due to the detachment.

6. Figure 2f is very difficult to interpret in terms of F-actin filaments causing nuclear deformations given the low resolution. It is claimed that Figure 2e shows how actomyosin fibers 'bisect the nucleus and cause nuclear fragments'. But in Figure 2e, at least a few examples can be seen where actin fibers are not bisecting or touching the nucleus. Are all deformations of the nucleus due to impinging actomyosin fibers? The paper does not put forth a convincing mechanism for how the nucleus is fragmenting upon knockdown of PPP1R12A and PPP1CB.

We do not claim that all actomyosin filaments in PPP1R12A-depleted cells are engaged in nuclear compression and fragmentation. Our data demonstrate that actomyosin activity is key to nuclear damage in myosin phosphatase-depleted cells. At nuclear compression sites proximal to nuclear envelope rupture sites, we do detect actomyosin filaments consistent with our hypothesis. These filaments are shown in **Fig 3c** and in the newly added **Supplementary Fig. 10c** of the revised manuscript. Despite the dynamic movement of nuclei in PPP1R12A-depleted cells, we find that 81.3% (165 out of 203) of nuclear compression and rupture sites were associated with actomyosin filaments in fixed cells – **these new data are now also included in Supplementary Fig. 10b**. In addition, we have performed physical micromanipulation experiments in collaboration with the Piel lab that showed that nuclear envelope rupture and chromatin extrusion, akin to the phenotypes observed in PPP1R12A-depleted cells, can be recapitulated by compressing nuclei (**Figure R3 below**). A movie of the compression experiments is uploaded for the benefit of the reviewer (**Nuclear compression.avi**). This shows that compressive force alone can trigger nuclear envelope rupture (the catastrophic mitosis at the end of the movie is interesting, but not pertinent to this work). Thus, our data strongly demonstrate that actomyosin filaments are associated with nuclear compression and nuclear envelope rupture sites and suggest that these filaments are causally related to the nuclear alteration. The correlative light and electron microscopy image analysis (cLEM) in **Figure S10b** of the revised manuscript shows a high-resolution image of actin filaments at a compression and rupture site.

Figure R3: Experimental setup (left panel). Frame from live cell recordings of mechanical compression of cells using a PDMS piston (arrows) (H2B, green; LAP2, red) (right panel). The live-cell movie of the compression experiment will be attached to these comments. Arrows indicate nuclear lamina rupture and chromatin extrusion sites.

7. The *in vivo* measurements are unconvincing. Nuclear abnormalities are not clearly visible in the picture provided (Figure 3c). It is not clear that the decrease in variance of the measured parameters necessarily indicates a smoother nucleus upon treatment with Y27632.

Also at the request of reviewer #3 we have clarified the z-section issue to more convincingly make this point (**Supplementary Figure S14a**). We also provide higher magnification *in vivo* images taken with a 63x 1.2NA objective – **now included in Supplementary Figure 14b of the revised manuscript; and in greater detail in Figure R4 on the following page**. Z-sections and separate channels are shown, with the arrow indicating a point of nuclear envelope rupture. These data can be included in the manuscript to address the reviewer's comment.

Figure R4: Intravital images of cells in a mammary tumors (MDA-MB-231) with H2B shown in red and LAP2B in green. Left panel shows different confocal z-sections, right panel shows single channels, * indicates the z-section split into single channels in the right-hand panels. Scale bar is 10 μ m.

Finally, the reviewer is correct that the variance measurements do not demonstrate smoothness, but this is not what we claim. We claim that the variance over time reflects the dynamic deformation of the nucleus. The overall changes in nuclear morphology are not as pronounced as *in vitro*; this, most likely, results from the constraints on shape conferred by the complex matrix geometry *in vivo*. This is not changed by short-term administration of Y-27632. Despite this, the metrics of nuclear morphology provided in **Supp. Fig. 13b** indicate an increase in circularity, with no significant change in aspect ratio, which is consistent with the nuclei becoming 'smoother' even though we do not go as far as making this statement.

8. Could the results in Figure 4 be interpreted in another way? Chromosomal spreads are done by colliding cells with a solid surface which forces the chromosomes apart. Knockdown of phosphatases makes the cytoplasm stiffer which might cause breaks in chromosomes.

The experimental protocol used for chromosome spreading including fixation and hypotonic swelling prior to spreading makes it very unlikely that chromosome breakage is caused by a

cytoplasm with higher stiffness. Also, it would not explain other chromosomal aberrations that we detected and show in **Figure 7** of the revised manuscript. Our analysis of control siRNA-transfected cells demonstrates that the procedure of spreading itself does not cause an elevated background of chromosomal aberrations. Crucially, following depletion of PPP1R12A we detect enhanced levels of the DNA damage markers γ H2AX and 53BP1 in fixed cells that were not subjected to the chromosome spreading procedure (**Figure 6**). The elevated γ H2AX signal and chromosomal aberrations detected in PPP1R12A-depleted cells are effectively suppressed by addition of the ROCK inhibitor Y-27632. For these reasons we are confident that actomyosin activity induces a genome instability phenomenon in cells lacking myosin phosphatase.

9.A direct comparison of actomyosin forces generated by the different cancer cell lines, along with a comparison of mechanical properties of the nucleus would be required along with evidence that myosin inhibition dynamically causes the highly abnormal nucleus to relax back to its 'smooth' shape.

So far we were not able to optimize hydrogel properties to conduct traction force microscopy (TFM) for the estimation of forces generated by cancer cells following myosin phosphatase depletion. It is also not clear whether forces impacting the nucleus in myosin phosphatase depleted cells would be transmitted to the substratum and read out by TFM. Nevertheless, it is known that myosin phosphatase antagonizes contractility and we show in **Figure 2a** and **Supplementary Fig 7a** that loss of PPP1R12A and PPP1CB increases the level of pMLC, a well-established marker for contractility. In **Figure 4b** of the revised manuscript, we now show the impact of PPP1R12-depletion in primary fibroblasts and carcinoma cells isolated from the same patient and refer to published work that the nuclei of transformed cells are softer. Since we do not directly measure forces in our manuscript, we will ensure that sentences involving the term force will be labeled as a hypothesis and not a firm conclusion/statement. To re-iterate the first part of this response, we have not been able to fully moderate the language in the resubmission because an editorial oversight meant that we did not receive the reviewer's comments until after we had re-submitted our work.

REVIEWERS' COMMENTS:

Reviewer #4 (Remarks to the Author):

The authors have addressed my comments satisfactorily with one exception.

The authors do not respond to the question on cell shape and nuclear shape. Treating with actin disruptors or myosin inhibitors can cause changes in cell shape, which affects nuclear shape. Language should be included that mentions this as a potentially complicating aspect that was not accounted for (the paper does not quantify cell shape). In this context, the authors should cite Versaevel et al, PMID 22334074 and Li et al, PMID 26287620 which show that the nuclear shape depends on cell shape, and inhibitors that cause changes in cell shape also change nuclear shape.

Response to Reviewers' comments

We would like to thank Reviewers #4 for his/her statement that "The authors have addressed my comments satisfactorily with one exception." In the revised version of our manuscript we have addressed this remaining point as detailed below.

Response to Reviewer #4

The authors have addressed my comments satisfactorily with one exception.

The authors do not respond to the question on cell shape and nuclear shape. Treating with actin disruptors or myosin inhibitors can cause changes in cell shape, which affects nuclear shape. Language should be included that mentions this as a potentially complicating aspect that was not accounted for (the paper does not quantify cell shape). In this context, the authors should cite Versaevel et al, PMID 22334074 and Li et al, PMID 26287620 which show that the nuclear shape depends on cell shape, and inhibitors that cause changes in cell shape also change nuclear shape.

We thank the reviewer for pointing this out. In the revised version of our manuscript, we have added a new paragraph to the discussion that addresses this point (page 13). The new section cites the two papers mentioned above and acknowledges the fact the altering or blocking actomyosin contractility also changes cell shape, which could contribute to the nuclear abnormalities detected in myosin phosphatase-depleted cells and to the rescue effect of actomyosin contractility inhibitors. In this new section, we also highlight data presented in our manuscript that strongly suggest that actomyosin fibers and contractility directly damage nuclear integrity in myosin phosphatase-depleted cells rather than indirectly via altering cell shape.